# Glioblastoma cells vampirize WNT from neurons and trigger a JNK/MMP signaling loop that enhances glioblastoma progression and neurodegeneration

Marta Portela[1,☉,¤,*], Varun Venkataramani[2,3,4], Natasha Fahey-Lozano[1], Esther Seco[1], Maria Losada-Perez[1], Frank Winkler[2,3], Sergio Casas-Tintó[1,☉,*]

**1** Instituto Cajal-CSIC, Madrid, Spain, **2** Neurology Clinic and National Center for Tumor Diseases, University Hospital Heidelberg, Heidelberg, Germany, **3** Clinical Cooperation Unit Neurooncology, German Cancer Consortium (DKTK), German Cancer Research Center (DKFZ), Heidelberg, Germany, **4** Institute for Anatomy and Cell Biology, Heidelberg University, Heidelberg, Germany

☉ These authors contributed equally to this work.
¤ Current address: Department of Biochemistry and Genetics, La Trobe Institute for Molecular Sciences, La Trobe University, Melbourne, Australia
* scasas@cajal.csic.es (SCT); m.portela@cajal.csic.es (MP)

**Data Availability Statement:** All relevant data are within the paper and its Supporting Information files.

## Abstract

Glioblastoma (GB) is the most lethal brain tumor, and Wingless (Wg)-related integration site (WNT) pathway activation in these tumors is associated with a poor prognosis. Clinically, the disease is characterized by progressive neurological deficits. However, whether these symptoms result from direct or indirect damage to neurons is still unresolved. Using *Drosophila* and primary xenografts as models of human GB, we describe, here, a mechanism that leads to activation of WNT signaling (Wg in *Drosophila*) in tumor cells. GB cells display a network of tumor microtubes (TMs) that enwrap neurons, accumulate Wg receptor Frizzled1 (Fz1), and, thereby, deplete Wg from neurons, causing neurodegeneration. We have defined this process as "vampirization." Furthermore, GB cells establish a positive feedback loop to promote their expansion, in which the Wg pathway activates cJun N-terminal kinase (JNK) in GB cells, and, in turn, JNK signaling leads to the post-transcriptional up-regulation and accumulation of matrix metalloproteinases (MMPs), which facilitate TMs' infiltration throughout the brain, TMs' network expansion, and further Wg depletion from neurons. Consequently, GB cells proliferate because of the activation of the Wg signaling target, β-catenin, and neurons degenerate because of Wg signaling extinction. Our findings reveal a molecular mechanism for TM production, infiltration, and maintenance that can explain both neuron-dependent tumor progression and also the neural decay associated with GB.

## Introduction

The progression of glioblastoma (GB) is accompanied by broad neurological dysfunctions, including neurocognitive disturbances that compromise quality of life during the short life

**Funding:** MP holds a fellowship from the Juan de la Cierva program IJCI-2014-19272 from the Spanish MICINN. Research has been funded by grant BFU2015-65685P. The funders had no role in study design, data collection and analysis, decision to publish, or preparation of the manuscript.

**Competing interests:** The authors have declared that no competing interests exist.

**Abbreviations:** APC, adenomatous polyposis coli; Arm, Armadillo; bGal, beta-galactosidase; Bsk^DN, dominant negative form of the effector Basket; Cyt-Arm, Cytoplasmic-Armadillo; dEGFRλ, constitutively active form of the Epidermal Growth Factor Receptor; dPI3K92E^caax, activated membrane-localized version of the PI3K catalytic subunit p110α/PI3K92E; ECM, extracellular matrix; EGFR, Epidermal Growth Factor Receptor; egr, eiger; ELAV, embryonic lethal abnormal vision; EST, Expression Sequence Tags; Fz, Frizzled; Fz1, Frizzled1; FZD, Frizzled; GAP43, Growth-Associated Protein-43; GB, Glioblastoma; GBSC, glioblastoma multiforme stem-like cell; GFP, green fluorescent protein; GFP-MLC, GFP tagged version of Myosin Light Chain protein; GMA-GFP, GFP tagged version of Moesin; GPI-YFP, glycosylphosphatidylinositol-anchored Yellow Fluorescent Protein; GRASP, GFP Reconstitution Across Synaptic Partners; Grnd, Grindelwald; grnd-extra, extracellular domain of Grnd; Hh, hedgehog; Hrp, horseradish peroxidase; HS, hybridization solution; HSP90B, Heat Shock Protein 90B; igl, igloo; Ihog, Interference hedgehog; JNK, cJun N-terminal kinase; LRP, lipoprotein-related protein; MMP, matrix metalloproteinase; myr, myristoilated; nkd-lacZ, transcriptional beta-galactosidase reporter of *naked* gene NLGN3, Neuroligin-3; NLS, nuclear localization signal; NMJ, neuromuscular junction; NPC, neural precursor cell; PI3K, phosphatidylinositol-3 kinase; PLA, proximity ligation assay; PP2A, protein phosphatase 2A; PTEN, Phosphatase and tensin homolog; qPCR, quantitative Polimerase Chain Reaction; RFP, Red Fluorescent Protein; Rho/ROCK, rhomboid/Rho associated kinase; RTK, receptor tyrosine kinase; SPARC/SPARCL1, Secreted Protein Acidic And Cysteine Rich; Sqh-GFP, GFP tagged form of spaghetti squash; SVZ, subventricular zone; TEM, transmission electron miscroscopy; TM, tumor microtube; TOR, Target of Rapamycin; TRE-RFP, RFP fluorescent protein regulated by a transcriptional response element of JNK pathway; TTHY1, Tweety homologue-1; Wg, Wingless; WNT, wingless-related integration site; WT, wild type.

span of affected patients, which is still around 1 year [1]. These tumors are often resistant to standard treatments, which include resection, radiotherapy, and chemotherapy with temozolomide [2]. Numerous studies are focused on new molecular targets to treat GBs [3–6]; however, none of them has yet proven effective, which is in stark contrast to the considerable progress made in many other tumor types. It is therefore necessary to better understand the key glioma-specific features of tumor biology that can lead to additional therapeutic strategies against GBs.

GB cells extend ultra-long membrane protrusions that interconnect tumor cells [7]. These TMs are associated with the worst prognosis in human gliomas. TMs contribute to invasion and proliferation, resulting in effective brain colonization by GB cells. Moreover, TMs constitute a multicellular network that connects GB cells over long distances (up to 500 μm in length), a feature that likely provides resistance against radiotherapy, chemotherapy, and surgery [7,8]. In addition, TMs seem akin to a basic mechanism of cell–cell connection and molecular communication called "cytonemes" in *Drosophila* [9]. *Growth-Associated Protein-43* (*GAP43*) is essential for the development of TMs and, thus, the tumor cell network associated with GB progression [7]. However, many aspects of this paradigmatic finding in glioma biology are still unexplored, including the molecular mechanisms underlying TM expansion and its impact on neighboring neurons.

Among signaling pathways, the most prominent candidate to play a role in GB progression is the WNT canonical pathway, which is activated upon the ligand "Wingless-related integration site" (WNT) binding to the specific family of receptors, low-density lipoprotein-related protein (LRPs) or Frizzled (FZD) in the plasma membrane. As a consequence of WNT binding, the "destruction complex" that includes the tumor suppressors Axin and adenomatous polyposis coli (APC), the Ser/Thr kinases GSK-3 and CK1, protein phosphatase 2A (PP2A), and the E3-ubiquitin ligase β-TrCP [10], is inactivated, and β-catenin (armadillo in *Drosophila*) accumulates. β-catenin translocates to the cell nucleus where it promotes the expression of target genes (i.e., *Cyclin D1* and *Myc*) that are important for cell proliferation [11,12]. The WNT pathway is evolutionarily conserved in all metazoans, and it plays a central role in brain development [13], adult neuronal physiology [14], and synaptogenesis [15,16]. Perturbations in WNT signaling are associated with neural deficits, Alzheimer disease [16–19], and, most notably, GB [20–22]. WNT/FZD signaling is frequently up-regulated in GB [23,24] (reviewed in the work by Suwala and colleagues [25]). In particular, one of the hallmarks of bad prognosis is the accumulation of ß-catenin in tumor cells [26,27], indicating an activation of WNT/FZD pathway [28].

Another signaling pathway that has been associated with glial proliferation is the cJun-N-terminal Kinase (JNK) pathway. GB cells normally activate the JNK pathway to maintain the stem-like state, and it has become a pharmacological target for the treatment of GB [29]. Moreover, the JNK pathway is the main regulator of several factors that regulate cell motility in different organisms and tissues [30–34]. Cell motility demands the remodeling of the extracellular matrix (ECM), which is mainly performed by a family of endopeptidases known as Matrix metalloproteinases (MMPs). There are more than 20 members, including collagenases, gelatinases, stromelysins, some elastases, and aggrecanases [35]. The vertebrate MMPs have overlapping substrates, they exhibit functional redundancy and compensation, and pharmacological inhibitors are nonspecific. In contrast, there are only 2 MMP genes in *Drosophila*, MMP1 and MMP2, categorized by their pericellular localization, with MMP1 being secreted and MMP2 being plasma membrane anchored. However, the products of both genes are found at the cell surface and released into media, the 2 *Drosophila* MMPs cleave different substrates, and GPI-anchored MMPs promote cell adhesion when they become inactive [36,37]. MMPs are up-regulated in a number of tumors, including gliomas [38]. MMP up-regulation

in GB is associated with the diffuse infiltrative growth, and they have been proposed to play a role in glioma cell migration and infiltration ([39,40] reviewed in the work by Nakada and colleagues [38]). Consequently, MMP up-regulation in GB is an indicator of poor prognosis [41], and therefore the study of the mechanisms mediated by MMPs is relevant for the biology of GB and cancer in general.

All in all, both the rapid growth of malignant gliomas and the neurological sequelae of affected patients are considerable challenges in neurooncology until today. A positive correlation between neuronal activity and glioma progression has recently been suggested [42,43]. The neuron-glioma interactions might, however, be manifold. Thus, the major aim of this study is to better characterize the complex world of interactions between neurons of the brain on one side and tumor cells on the other. The results can provide a framework of a novel understanding of the disease.

## Results

### *Drosophila* glioma network

To study the mechanisms of communication among malignant glial cells with neighboring neurons, we used a previously described *Drosophila* GB model [44], which involves the co-overexpression of constitutively active forms of Epidermal Growth Factor Receptor (dEGFRλ) and an activated membrane-localized version of the PI3K catalytic subunit p110α/PI3K92E (dPI3K92E$^{caax}$) under Gal4 *UAS* control, specifically driven in the glial cells by means of *repo-Gal4* [45,46]. This combination stimulates malignant transformation of postembryonic larval glia, leading to lethal glial neoplasia [44,47], which is measured by the increase of glial cell number (identified by the green fluorescent protein reporter [GFP$^{NLS}$]) compared with a control brain (Fig 1A–1C). Using this *Drosophila* GB model, we have evaluated the impact of glial tumor cell increase on neighboring neurons.

To visualize the total volume of the glial plasma membrane in larval brains, we expressed a myristoylated form of red fluorescent protein (expressed via *UAS-myrRFP*). Red fluorescent protein (RFP) signal in control brains shows the glial membrane and the network formed among wild-type (WT) glial cells (Fig 1D) and a significant enlargement in glial membrane volume of glioma brains (Fig 1E).

A detailed analysis of the contact region between neuron and GB cells revealed that glial protrusions wrap clusters of neurons in individual GB "nests" (Fig 1F and 1G and see S1 and S2 Videos); this organization is comparable to previously described perineuronal nests in GB patients [48]. To confirm the neuronal identity of the cells within the nests, we used neuron specific antibodies (anti- horseradish peroxidase (Hrp) and anti- embryonic lethal abnormal vision (ELAV), green in Fig 1F and 1G) and the Interference hedgehog (Ihog)-RFP or myristoilated (myr)-RFP reporters to visualize the TMs. The results show that TMs infiltrate and enwrap neurons, thereby segregating neuronal populations.

In addition, we obtained transmission electron microscopy (TEM) images to visualize glial morphology in control samples (Fig 1H and 1I). High magnification TEM images show an enlargement of glial cell surfaces in glioma brain samples (Fig 1K), and the infiltration of glioma cell projections through the brain (Fig 1L). Glioblastoma TMs are described in human GB samples as a cell to cell communication system [7], and we found here comparable structures in *Drosophila* glioma brain samples. Magnifications of TEM images show the details of perineuronal nests of glial protrusions that surround and isolate neurons (Fig 1M and 1N, illustration in J).

TMs share characteristics with the cytonemes previously described in *Drosophila* epithelial cells [7]. Ihog (Interference hedgehog) is a type 1 membrane protein shown to mediate the

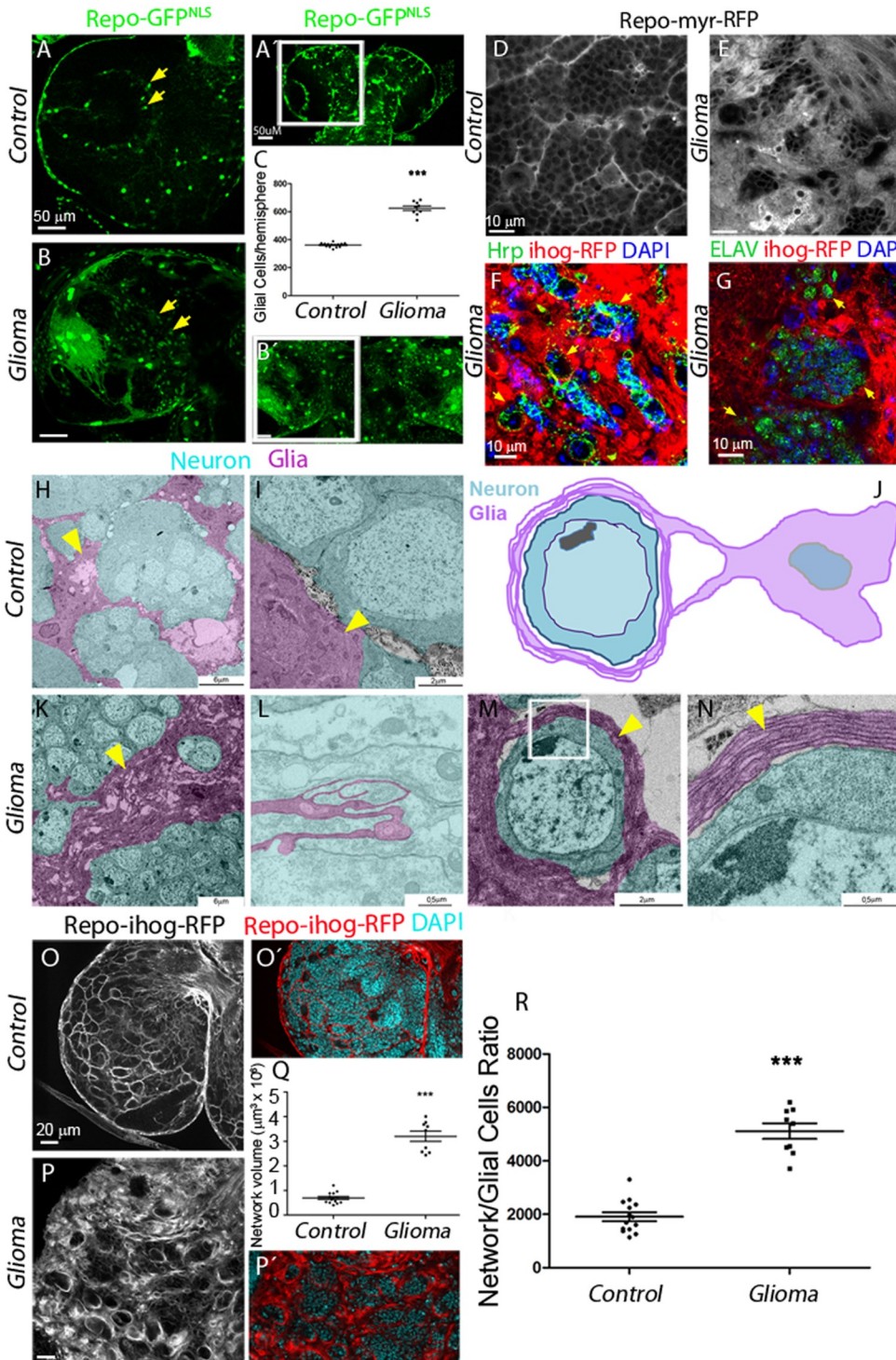

**Fig 1. Co-activation of EGFR-Ras and PI3K in *Drosophila* glia causes an expansion of the glial network.** Brains from third instar larvae. Glia nuclei are labeled with GFP$^{NLS}$ (green) driven by repo-Gal4. Each brain is composed of 2 symmetrical hemispheres. (A, B, A′, B′) In glioma brains (repo>dEGFR$^λ$; dp110$^{CAAX}$) (B), both brain hemispheres are enlarged, and the number of glial cells is increased relative to WT control (A–A′). Higher magnifications are shown in panels A and B. The quantification of the number of glial cells is shown in (C). Arrows indicate glial nuclei. (D–E) Optical sections of larval brain to visualize the glial network, glial cell bodies, and membranes are labeled (myrRFP). (D) RFP signal (in grayscale) in control brains shows glial membranes and the network in WT brain. (E) The glioma brain shows a dramatic increase in the membrane projections and in the size of the network. (F) Glia is

labeled with UAS-Ihog-RFP (red) driven by repo-Gal4 to visualize TMs in glioma cells; nuclei are marked with DAPI (blue) and neurons are stained with Hrp (green) and enwrapped by glial TMs in glioma brains (yellow arrowheads). (G) Glia is labeled with UAS-myr-RFP driven by repo-Gal4 to visualize membrane projections in glial cells (red); nuclei are marked with DAPI (blue) and neurons are stained with ELAV (green) and enwrapped by glial TMs in glioma brains (yellow arrows). (H–I and K–L) TEM images of third instar larval brains expressing HRP in the glial cells. (H–I) HRP deposits label cell membranes in dark, thus identifying glial cells. Colored images from control brains show glial cells identified by HRP staining (magenta) and HRP-negative neurons (cyan). (J) Schematic diagram: a glioma cell labeled with HRP (magenta) showing that glioma cells produce a network of TMs that grow to surround neighboring neurons (cyan). (K–N) Several magnifications of glioma brains showing TMs that grow and enwrap neighboring neurons (cyan). Detail of several layers of a glioma membrane enwrapping a neuron (M–N) and a longitudinal section of a TM, showing to glioma cells (black arrows) connected by TMs (L), yellow arrows indicate glial membranes. (O–R) Glia are labeled with UAS-ihog-RFP to visualize active cytoneme/TM structures in glial cells as part of an interconnecting network. (O) In control brains, the active glial cytonemes are shown by repo>ihog-RFP in gray or red in the merge. In glioma brains (P), the TMs grow and expand across the brain; quantification of the network volume (Q) and the network/glial cell ratio is shown in panel R. Error bars show SD; ***$P < 0.0001$. Scale bar size is indicated in this and all figures. The numerical data pertaining to this figure can be found in S1 Data file. Genotypes: (A) w; repo-Gal4, UAS-GFPnls/UAS-lacZ, (B) UAS-dEGFRλ, UAS-dp110CAAX; repo-Gal4, UAS-GFPnls, (D) w; Gal80ts; repo-Gal4, UAS-myrRFP/UAS-lacZ, (E, G) UAS-dEGFRλ, UAS-dp110CAAX; Gal80ts; repo-Gal4, UAS-myrRFP, (F, P) UAS-dEGFRλ, UAS-dp110CAAX; repo-Gal4, UAS-ihog-RFP, (H-I) w; UAS-HRP:CD2; repo-Gal4/UAS-lacZ, (K-N) UAS-dEGFRλ, UAS-dp110CAAX; UAS-HRP:CD2; repo-Gal4, (O) w; repo-Gal4, UAS-ihog-RFP/UAS-lacZ. EGFR, Epidermal Growth Factor Receptor; ELAV, embryonic lethal abnormal vision; GFP, green fluorescent protein; Hrp, horseradish peroxidase; PI3K, phosphatidylinositol-3 kinase; RFP, Red Fluorescent Protein; TEM, transmission electron microscopy; TM, tumor microtube; wt, wild type.

response to the active Hedgehog (Hh) protein signal [49]. Ihog accumulates in the epithelial cytonemes, and it is commonly used to visualize these structures [50,51]. To visualize glial projections in the entire brain of *Drosophila*, we expressed a *UAS-Ihog-RFP* construct under the control of *repo-Gal4*. This red fluorescent tagged form of Ihog-RFP in epithelial cells labels cellular processes (cytonemes) in the basal region of the wing imaginal discs [50,52]. The expression of *UAS-Ihog-RFP* under the control of *repo-Gal4* allows the visualization of projections in WT glia that build an interconnecting network (Fig 1O). The accumulation of Ihog in transformed glial cells allowed visualizing the TM-like processes (Fig 1P). We used 3D reconstructions (IMARIS Bitplane, see Materials and methods) from confocal stacks of images to quantify the volume of glial membrane projections, and we observed an expansion of the TM-like processes in glioma compared with control brains (Fig 1Q).

To determine whether TM-like processes expand as a consequence of the increase in the number of GB cells, we quantified the volume of the TM-like processes and divided by the number of glial cells (Fig 1R). The results show that TM-like processes volume with respect to the glial cell number ratio is higher in GB, and therefore we conclude that TM-like processes expand in GB beyond the increase in the number of glial cells.

TMs were previously described to be actin-based projections (as cytonemes) [7]. To further determine the identity of the TM-like processes in *Drosophila* glioma cells, we took advantage of the LifeActin-GFP reporter and observed that the TM-like processes (Ihog positive) are also actin-rich (S1A–S1C Fig). Moreover, confocal 3D reconstructions with the glial network marked with Ihog-RFP and LifeAct-GFP showed that actin-based TM-like processes form perineuronal nests (S1A and S1B Fig and S4 and S5 Videos). Additionally, we characterized the TM-like processes with 4 previously described markers for cytonemes in *Drosophila* [52] and orthologue markers of human TMs [7,52] (GFP tagged version of Moesin [GMA-GFP], glycosylphosphatidylinositol-anchored Yellow Fluorescent Protein [GPI-YFP], GFP tagged version of Myosin Light Chain protein [GFP-MLC], and GFP tagged form of spaghetti squash [Sqh-GFP] in S1D–S1G Fig). Moreover, using methods to prevent epithelial cytoneme formation (by down-regulating *neuroglian* [*nrg-RNAi*]) [53], we found that the TM-like processes in GB were also reduced (S1H and S1J Fig). These results suggest that glioma cells build and organize TM networks around the neurons. Because of the similarity of *Drosophila* GB cell

protrusions to those observed in patients with glioma and because the term TMs is now the established terminology to describe thin membrane protrusions from malignant glioma cells in human and murine tumors [7], we use this term from here on to describe the membrane protrusions observed in GB *Drosophila* cells.

Next, we sought to clarify whether similar molecular machineries are involved in human and *Drosophila* TMs. Growth-Associated Protein 43 (GAP43) is necessary for TM formation and function, and it drives microtube-dependent tumor cell invasion, proliferation, interconnection, and radioresistance [7]. To determine whether the *Drosophila* glial network is susceptible to *GAP43* depletion, as it has been described in human tumor cells [7], we knocked down *igloo* (*igl*), the invertebrate *GAP43*-like gene [54], in glioma cells. Larval brain images show that the glioma network does not develop upon *igl* silencing, and, as a consequence, glial TMs no longer enwrap neurons (S1L and S1M Fig and S1–S3 Videos). We excluded the possibility that suppression of TM expansion and glioma progression was due to a titration of GAL4 activity caused by introducing an additional UAS-transgene (*UAS-lacZ* or *UAS-yellow-RNAi*; S2A and S2B Fig).

The direct consequence on flies developing a glioma is larval/pupal lethality. Upon *igl* knockdown, however, the glioma-induced lethality is prevented, allowing the emergence of adults (S1N Fig). Interestingly, *igl* knockdown in WT glial cells neither affects the normal development of neurons and glia nor their viability (S2C–S2G Fig). Taking all data together, transformed glial cells take advantage of the *igl*/*GAP43*-dependent tumoral network to proliferate and enwrap neurons and, as a consequence, cause larval/pupal death. These observations indicate that *Drosophila* and human gliomas share the *igl*/*GAP43* requirement, and thus, the TM network is structurally similar in both species.

## WNT/Wingless signaling in glioma

WNT signaling has long been suggested as a hallmark in gliomagenesis associated with the proliferation of stem-like cells in human GBs [55] and resistance to chemotherapy and radiotherapy (reviewed in the work by Suwala and colleagues and the work by Paw and colleagues [25,56]). However, there is no correlation between overexpression of WNT ligands and GB development (S3 Fig).

β-catenin activation (translocation to the nuclei) is a downstream event of WNT pathway activation. It has been identified in 19% of adult and in 30% of pediatric patient glioma samples [48]. Moreover, WNT pathway inhibition leads to suppression of tumor growth, cell proliferation in cultures, and a modest induction of cell death [55].

Thus, we assessed in parallel *Wingless* (*wg*) expression, the fly homologue of human WNT, and Wg pathway activation in the glioma model. We quantified Wg levels (using an anti-Wg antibody) in glial cells and neuronal tissue. The results showed that whereas Wg protein is homogeneously distributed in control brains with a slight increase (1:5 Glia/Neuron ratio) in glial membranes (Fig 2A and 2C), glioma brains showed a 4-fold increase of Wg Glia/Neuron ratio (Fig 2B and 2C), in line with WNT accumulation in human GBs [57,58]. To determine whether Wg could be signaling to the glial cells, we assessed the presence of Frizzled (Fz) receptors in glial membranes. A specific monoclonal antibody was used to visualize the Fz1 receptor, and the quantification of anti-Fz1 signal in glia versus neurons showed that Fz1 receptors are localized homogeneously across the brain with a relative accumulation in glial membranes in control samples (Fz1, Fig 2D and 2H). However, in the glioma model, the accumulation of Fz1 protein is increased in the glioma cells (Fig 2E and 2H), similar to Wg. Therefore, we raised the working hypothesis that the TM network contributes to Wg/Fz1 signaling in glioma cells.

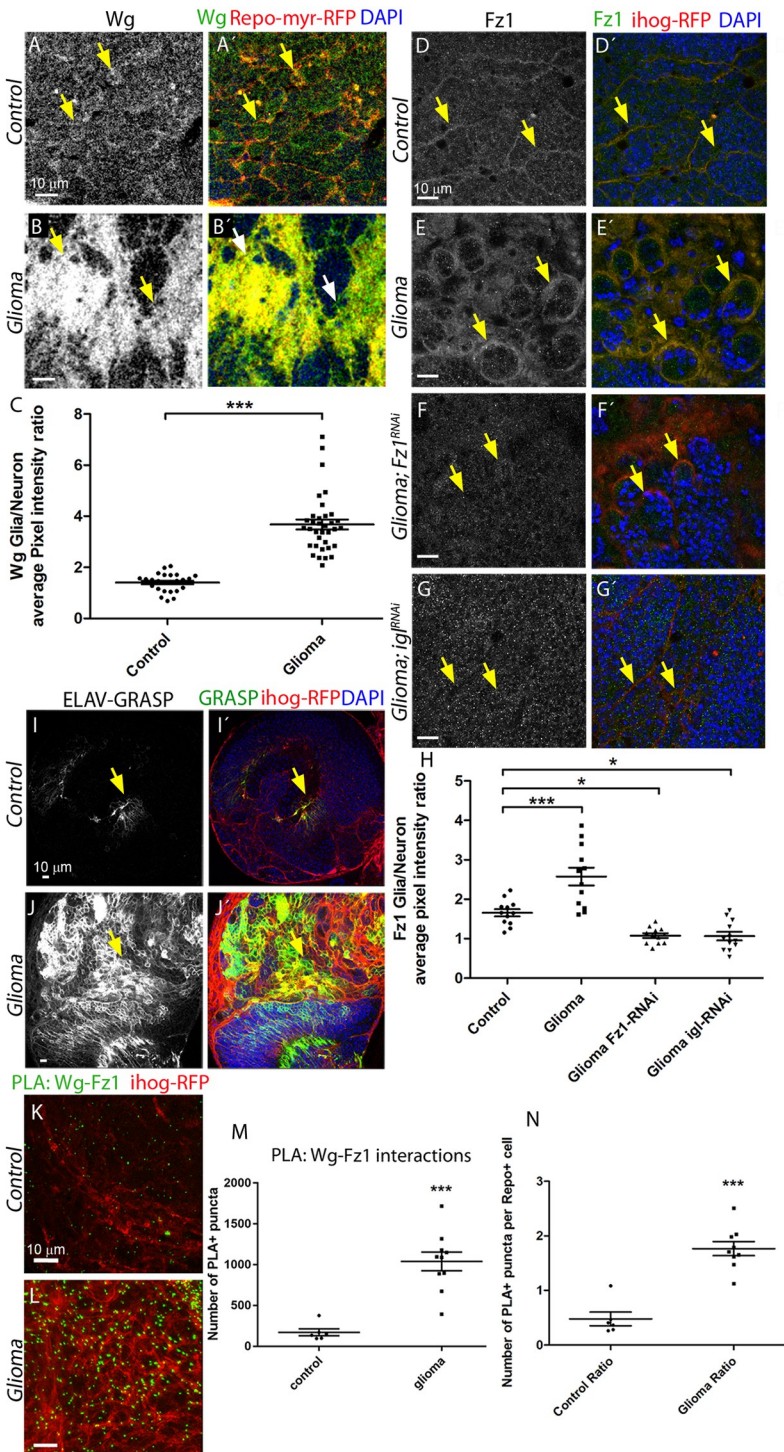

**Fig 2. Wg/Fz1 accumulate in glioma cells, and Fz1 in glia interacts with neuronal Wg.** Larval brain sections with glial cell bodies and membranes labeled in red (myrRFP) and stained with Wg antibody show homogeneous expression in the control brains (A) in gray (green in the merge). In the glioma brains, Wg accumulates in the glial transformed cells (B); the Wg average pixel intensity ratio between Glia/Neuron quantification is shown in panel C. Arrows indicate Wg staining in glial membranes. (D–G) Glial cells are labeled with UAS-Ihog-RFP to visualize the glial network (red) and stained with Fz1 (gray or green in the merge). (D) Fz1 is homogeneously distributed in control brains, with a slight accumulation in the Ihog+ structures. (E) Fz1 accumulates in the TMs and specifically in the projections that are in contact with the neuronal clusters. (F) Upon knockdown of fz1 in glioma brains, the tumoral

glial network is still formed but Fz1 is not detectable. (G) Knockdown of igl in glioma brains restores a normal glial network, and Fz1 shows a homogeneous distribution along the brain section. Arrows indicate Fz1 staining in glial membranes. (H) Fz1 average pixel intensity ratio between Glia/Neuron quantification. Arrows indicate Fz1 staining in glial membranes. (I–J) GRASP technique was used, and both halves of GFP tagged with a CD4 signal to direct it to the membranes (CD4-spGFP) were expressed in neurons (elav-lexA) and glial (repo-Gal4) cells, respectively. Only upon intimate contact is GFP protein reconstituted and green fluorescent signal is visible. (I) Control brains showed a discrete GFP (green) signal corresponding to the physiological interaction between glia and neurons. (J) In glioma brains, a strong GFP signal from the GRASP reporter is detected. Arrows indicate GRASP reconstitution GFP signal. (K–L) PLAs were performed in control and glioma brains to quantify the interactions between Wg and Fz1. (K) Control brains showed a discrete number of PLA+ puncta (green) showing the physiological interactions. (L) Glioma brains showed a 5-fold increase in the number of puncta, quantified in panel M. The number of PLA+ Wg-Fz1 interactions per Repo+ cell in control and glioma brains is shown in panel N. Arrows indicate PLA+ puncta. Nuclei are marked with DAPI (blue). Error bars show SD; $^{*}P < 0.01$; $^{***}P < 0.0001$. The data underlying this figure can be found in S1 Data. Genotypes: (A) w; Gal80ts; repo-Gal4, UAS-myrRFP/UAS-lacZ, (B) UAS-dEGFRλ, UAS-dp110CAAX; Gal80ts; repo-Gal4, UAS-myrRFP, (C) w;; repo-Gal4, ihog-RFP/UAS-lacZ, (D) UAS-dEGFRλ, UAS-dp110CAAX;; repo-Gal4, UAS-ihog-RFP, (E) UAS-dEGFRλ, UAS-dp110CAAX; UAS-Fz1-RNAi; repo-Gal4, UAS-ihog-RFP, (F) UAS-dEGFRλ, UAS-dp110CAAX;; repo-Gal4, UAS-ihog-RFP /UAS-igl-RNAi, (I) w;; elav-lexA; repo-Gal4, UAS-ihog-RFP/UAS-CD4-spGFP1-10, lexAop-CD4-spGFP11, (J) UAS-dEGFRλ, UAS-dp110CAAX; elav-lexA; repo-Gal4, UAS-ihog-RFP/ UAS-CD4-spGFP1-10, lexAop-CD4-spGFP11, (K) w;; repo-Gal4, ihog-RFP/UAS-lacZ, (L) UAS-dEGFRλ, UAS-dp110CAAX;; repo-Gal4, UAS-ihog-RFP. elav, embryonic lethal abnormal vision; Fz1, Frizzled1; GFP, green fluorescent protein; GRASP, GFP Reconstitution Across Synaptic Partners; igl, igloo; myrRFP, myristoilated Red Fluorescent Protein; PLA, proximity ligation assay; TM, tumor microtube; Wg, wingless.

To evaluate the contribution of Fz1 to the progression of glioma, we knocked down Fz1 receptor (using *UAS-Fz1-RNAi*) in transformed glial cells. Glioma brains showed a significant reduction of Fz1 protein (Fig 2F and 2H). However, the TM network was reduced although still formed (Fig 2F'). Furthermore, we prevented the development of the glioma network by expressing *UAS-igl-RNAi* and stained the brains for Fz1. Under these conditions, the network did not expand, Fz1 did not accumulate in TMs, and Fz1 showed a homogeneous distribution in glia and neurons (Fig 2G and 2H). These results indicate that Fz1 accumulation in TMs is a consequence of the glioma network.

The abnormal distribution of Wg and Fz1 in glioma brains could be due to either an increase in gene and/or protein expression or to redistribution of the proteins. The expression of both genes, as determined by quantitative Polimerase Chain Reaction (qPCR), is comparable in control and gliomas (S4A Fig). To consider whether Fz1 was regulated at the level of mRNA translation or protein stability and degradation, we measured total Fz1 protein levels by Western blots (S4B Fig). Quantification of the membranes showed no significant changes for Fz1 in glioma (S4B Fig). Moreover, in situ hybridization to detect Wg and Fz1 mRNA expression showed no differences for *wg* or *Fz1* transcription between controls and gliomas (S4C Fig).

Thus, in spite of a higher signal for Wg and Fz1 proteins in glioma cells, as detected by immunofluorescence, there are no changes in gene expression and total protein content. We further hypothesized that Fz1 is transported and accumulated along glioma TMs, which contact neighboring neurons and could receive Wg from them.

To assess this, we performed GFP Reconstitution Across Synaptic Partners (GRASP) experiments [59]. This technique determines physical interaction between glia and neurons, in the range of 20 to 40 nm (synaptic distance), which is compatible with protein-protein interaction [59] (see details in Materials and methods). Control samples (Fig 2I) showed a discrete signal corresponding to the physiological interaction between glia and neurons; however, upon glioma induction, a strong GFP signal was detected (Fig 2J). This result indicates that, in a glioma condition, there is a significant increase of glia-neuron membrane interaction, consistent with the TEM images (Fig 1K–1N), which showed nests of TMs surrounding and isolating neurons.

To confirm the physical interaction of Fz1 and Wg proteins, we performed proximity ligation assays (PLAs; see Materials and methods) [60]. This quantitative method reports the interactions between 2 proteins with a resolution of 40 nm [61]. Control brains showed a discrete number of puncta, indicative of interaction (Fig 2K and quantified in Fig 2M). However, glioma brains showed a 5-fold increase in the number of PLA-positive puncta (Fig 2L and 2M), indicating a significant increase in the number of Fz1-Wg interactions. Moreover, we calculated the number of Wg-Fz1 interactions per glial cell in control and glioma brains, and the results show that the number of interactions per cell is higher in the glioma brains (Fig 2N). These results confirm that glioma cells accumulate Fz1 receptor in TMs, and then this receptor interacts with Wg. Because Wg is not up-regulated in the glioma brain, it is possible that Wg originates from neighboring neurons, and it relocates and accumulates in glioma cell membranes.

## Wg/Fz1 pathway is active in glioma and inactive in neurons

Wg targets are indicators of Wg/Fz1 activity in the recipient cell. Armadillo/β-Catenin is a cytoplasmic protein which, upon activation of Wg pathway, translocates into the nucleus and activates transcription of target genes [20,62]. To determine whether Fz1 is signaling in gliomas as a consequence of Wg-Fz1 interaction, we used an anti-Armadillo (Arm) antibody, which identifies its cytoplasmic inactive form (Cyt-Arm) [63]. Cyt-Arm was homogeneously distributed along the brain in control samples (neuron/glia ratio = 1, Fig 3A and 3D). However, glioma brains showed accumulation of Cyt-Arm in cells neighboring gliomas (including neurons) and reduction in glioma cells (Fig 3B and 3D). This result suggests that, in glioma brains, the Wg/Fz1 pathway is inactive in healthy brain tissue and active in glioma cells. More importantly, these data show that the network expansion and the accumulation of Fz1 in the TM projections might have an effect on neighboring neurons. We prevented the formation of the glioma network expressing *UAS-igl-RNAi* or down-regulated Fz1 (*UAS-Fz1-RNAi*) and stained for Cyt-Arm. Under these conditions, Cyt-Arm was homogeneously distributed within the brain similar to the control (Fig 3C and 3D and S5A Fig), demonstrating that the TM network is required to promote Wg signaling in the transformed glia.

To further confirm that Wg pathway is active in the transformed glial cells and silent in healthy brain tissues, we used 5 additional Wg pathway reporters, namely, *arm-GFP* (which shows nuclear GFP when Wg signaling is active), *naked (nkd)-lacZ, teashirt (tsh)-lacZ, fz4-GFP*, and *dally-LacZ* [64,65]. All these reporters were active in glioma cells and inactive in neighboring cells (which includes neurons) compared to control brains (Fig 3E–3H and S5B–S5G Fig), thereby confirming previous results.

Because these *Drosophila* observations on the relative localization and accumulation of Wg and Fz1, and the Wg signaling reporters in the glioma cells are novel features of GB, we assessed the conservation of this mechanism in human samples. To that end, we used a primary patient-derived glioblastoma culture xenograft model using S24 cells kept under stem-like conditions (see Materials and methods), which reproduce previously described Scherer modes, perivascular migration and spread [48,66].

We stained the samples for β-catenin and WNT1 proteins in S24 xenograft brain sections and compared them to control samples (Fig 3I–3N and S6G and S6H Fig). We observed a significant increase of both WNT1 and nuclear β-catenin proteins in GB cells, in line with our observations in *Drosophila*. Thus, WNT1 accumulates on GB cells and activates the WNT pathway; consequently, β-Catenin is up-regulated leading to GB cell malignancy [27]. Moreover, WNT1 and β-catenin staining of grade II and III GB samples from S24 xenograft brain

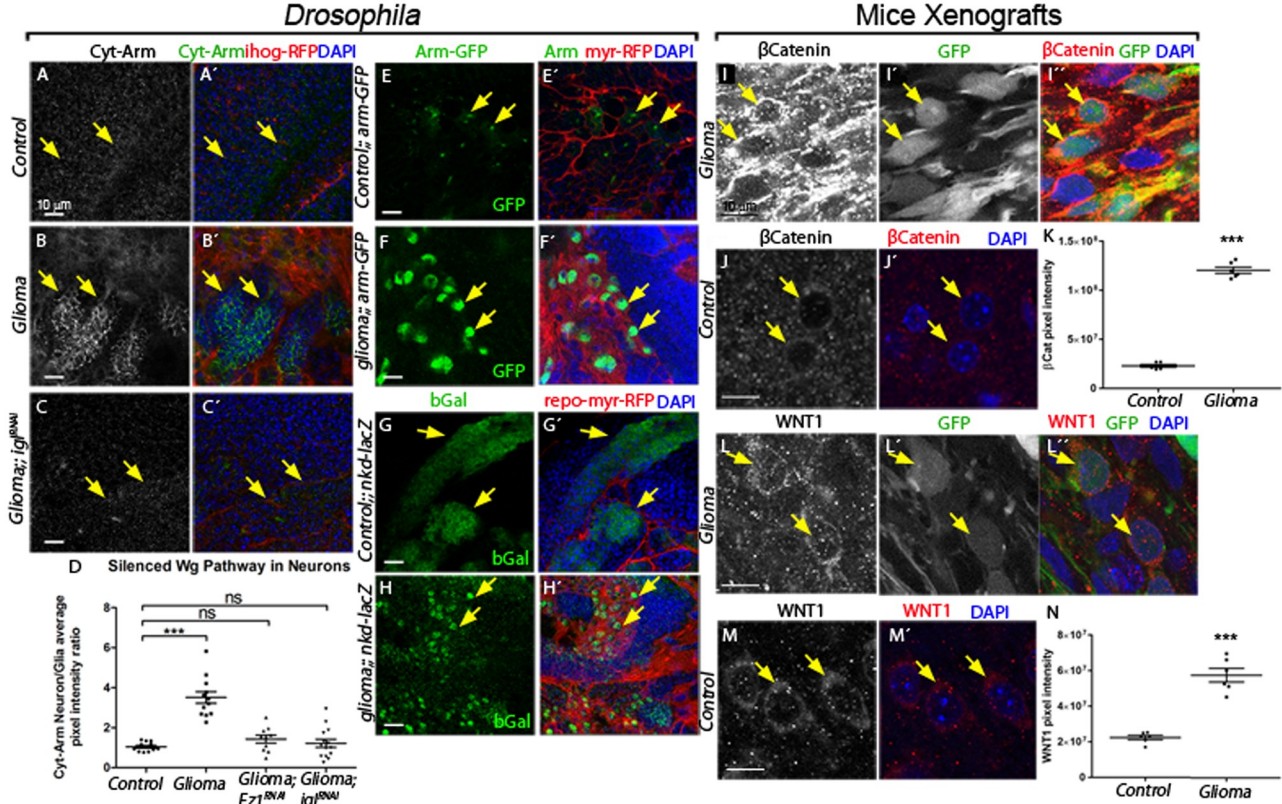

**Fig 3. Wg signaling pathway is active in glioma cells, and the glioma inactivates it in neuronal clusters in glioma brains.** (A–C) Larval brain sections with glial membrane projections labeled in red and stained with Arm (gray or green in the merge). (A) Cyt-Arm is homogeneously distributed in control sections. (B) In glioma brains, Cyt-Arm accumulates in the neurons' cytoplasm where it is inactive. (C) Knockdown of igl in glioma brains restores a normal glial network, and Cyt-Arm does not accumulate showing a homogeneous distribution similar to the control. Arrows indicate Cyt-Arm staining at the glia-neuron interphase. (D) Cyt-Arm average pixel intensity ratio between neuron/glia quantification showing the Wg signaling pathway silencing in neurons in a glioma brain. (E–H) Glial cell bodies and membranes are labeled with myrRFP (red). Wg signaling pathway reporters arm-GFP (E–F) and nkd-lacZ (stained with anti-bGal) (G–H) in control and glioma brains show activation of the pathway (green) in glioma cells (red) compared with the reporter activation mostly in neurons in the control brains. Arrows indicate cells with reporter activation. (I–N) Confocal immunofluorescence single plane images of S24 glioblastoma multiforme stem-like cells (GBSC) NMRI nude mice brains (glioma) and NMRI nude mice (control) brains stained with human anti-βCatenin (I–J) and WNT1 (L–M) both show in gray (red in the merged image) an increase in the glioma samples. The corresponding quantification of the pixel intensity is shown in panels K and N. Green signal from tumor cell GFP expression allows specific detection of S24 GBSC–related structures in the mouse brain (I′, L′). Arrows indicate glioma or control cells. Nuclei are marked with DAPI. Error bars show SD; ***$P < 0.0001$ and ns for nonsignificant. The data underlying this figure can be found in S1 Data. Genotypes: (A) w; repo-Gal4, ihog-RFP/UAS-lacZ, (B) UAS-dEGFRλ, UAS-dp110CAAX;; repo-Gal4, UAS-ihog-RFP, (C) UAS-dEGFRλ, UAS-dp110CAAX;; repo-Gal4, UAS-ihog-RFP /UAS-igl-RNAi, (E) w; Gal80ts; repo-Gal4, UAS-myrRFP/ arm-GFP, (F) UAS-dEGFRλ, UAS-dp110CAAX; Gal80ts; repo-Gal4, UAS-myrRFP/ arm-GFP, (G) w; Gal80ts; repo-Gal4, UAS-myrRFP/nkd04869a-lacZ, (H) UAS-dEGFRλ, UAS-dp110CAAX; Gal80ts; repo-Gal4, UAS-myrRFP/ nkd04869a-lacZ. Arm, Armadillo; bGal, beta-galactosidase; Cyt-Arm, Cytoplasm-Armadillo; GBSC, glioblastoma multiforme stem-like cell; GFP, green fluorescent protein; igl, igloo; myrRFP, myristoilated Red Fluorescent Protein; nkd-lacZ, nkd-lacZ, transcriptional beta-galactosidase reporter of *naked* gene NLGN3; NMRI, Naval Medical Research Institute.

sections and quantification of the pixel intensity indicated that the accumulation of WNT1 and activation of β-catenin correlates with the grade of the tumor (S6A–S6F Fig).

To determine the contribution of the Wg receptor Fz1 to the proliferation of *Drosophila* gliomas, we quantified the number of transformed glial cells upon *Fz1* down-regulation. The data showed a significant increase of glial cell number in glioma brains, which is prevented by *Fz1* down-regulation, as well as by TM network dismantlement upon *igl* knockdown (S7A–S7E Fig). These cellular features are reflected in an increase in adult viability upon *Fz1* or *igl* knockdown (S7F Fig). Moreover, triggering the GB condition in the adult reproduces Fz1 accumulation in GB cells and Wg pathway activation, indicating that the abnormal Wg/Fz1

signaling in GB is not dependent on the developmental stage (S7G–S7O Fig). In conclusion, our results show that Fz1 receptor expression is necessary in glioma cells to increase the number of glioma cells and induce the associated lethality in *Drosophila* adults.

## Glioma-relevant Wg originates in neurons

To determine whether the source of Wg is neuronal or glial, we silenced *wg* expression in neurons or in glioma cells separately in GB animals. Pan-neuronal *wg* silencing (*elav>wg-RNAi*) is lethal, consistent with the requirement of Wg in neuronal biology [15,67]. On the other hand, *wg* knockdown (*wg-RNAi*) in glioma cells (*Repo>dp110CAAX; EGFRλ; wg-RNAi*) does not prevent glioma cell proliferation (Fig 4A–4C and 4E) nor glioma TM expansion (Fig 4C′ and 4F). These results suggest that *wg* expression in glioma cells is not relevant for glioma progression in the fly model.

To further emphasize the contribution of glial Fz1 receptor as mediator for neuronal-Wg depletion, we generated glioma cells while silencing *Fz1* expression, and, in addition, we co-expressed a constitutively active form of *armadillo* (*UAS-armS10*) [68], which activates Wg pathway downstream Wg-Fz1 in these glioma cells (Fig 4D). To confirm that the Fz1 depletion and Arm signaling reproduces a glioma-like condition, we quantified the number of glial cells and TM network volume (Fig 4D–4F). The results show that the activation of Wg pathway (*arm$^{S10}$*) is sufficient to overcome *Fz1* knockdown in glioma cells and induce TMs expansion and glial cells proliferation (Fig 4E and 4F), similar to the control glioma (Fig 4B).

## Glioma depletes Wg from neuronal membranes

The mechanisms of Wg delivery are currently under debate. This protein was initially described as a secreted protein [69]. Recent studies have proved that Wg secretion is not necessary for *Drosophila* development [70]. A membrane-tethered version of Wg protein [71] (*Wg$^{NRT}$*) can substitute the endogenous gene, in mimicking Wg's normal functions and produce viable organisms [70]. We took advantage of this tool to determine the cellular mechanisms mediating glioma Wg retrieval from neurons. We created a genetic combination to substitute 1 copy of endogenous *wg* with 1 copy of *wg$^{NRT}$* exclusively in neurons (to avoid crossed expression, we used a different genetic driver system, the LexA-LexAop system [72]), while inducing a glioma marked with Ihog-RFP (by using the Gal4/*UAS* system). In addition, upon LexA system activation, neurons are marked with membranous GFP (CD8-GFP), whereas the rest of the cells are WT. We first analyzed the interaction of glioma cells with *wg$^{NRT}$*-expressing neurons (Fig 4G–4I). The results show that heterozygous *wg$^{NRT}$/wg* in neurons, prevented glioma network expansion (Fig 4I compared to Fig 4H, quantified in Fig 4J). Wg$^{NRT}$ is anchored to neuronal membranes; thus it would be expected to reduce the total Wg signaling in glioma cells thereby decreasing cell proliferation/survival and, consequently, resulting in a normal sized brain. In conclusion, glioma cells produce a network of TMs that reach neighboring neurons, increasing intimate membrane contact that facilitates neuronal-Wg sequestering (which we refer to as vampirization) mediated by the Fz1 receptor in glioma TMs. Moreover, Wg/Fz1 signaling in glioma mediates glioma cell proliferation and tumor progression.

## Gliomas cause neurodegeneration

Observations in patients document that neurodegeneration occurs in glioma brains (reviewed in the work by Savaskan and colleagues [73]). To measure neurodegeneration in our fly model, we quantified the number of active zones (synapses) in the neuromuscular junction (NMJ). NMJs quantification of confocal images stained with anti-bruchpilot (Nc82) revealed a

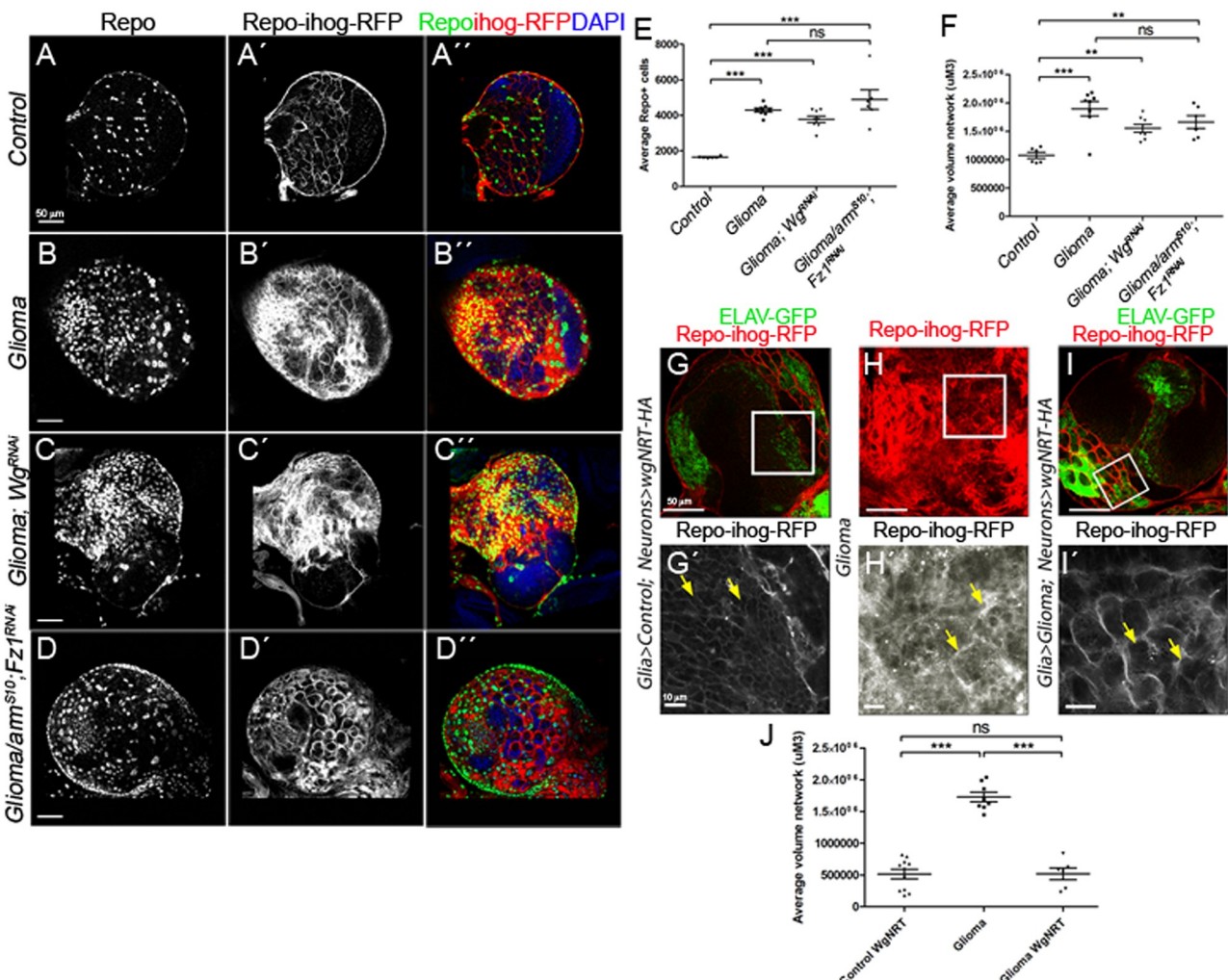

**Fig 4. Wg expression in glioma cells is dispensable for tumor progression because glioma depletes Wg from neuronal membrane.** (A–F) Larval brain sections with glial membrane projections labeled in gray (red in the merge) and glial cell nuclei stained with Repo (gray, green in the merge). (C) wg knockdown in glioma cells (wg-RNAi) or armS10; Fz1-RNAi (D) does not prevent glioma cell number increase nor glioma TM volume expansion quantified in (E–F). (G–G′) Control brains express 1 copy of WgNRT-HA instead of endogenous secretable Wg in neurons (green, ELAV-GFP). (H–H′) Glioma samples that do not express WgNRT in neurons show an overgrown TM network. (panel I in red, magnification in panel I′ in gray) Glioma brains expressing membrane-anchored Wg (WgNRT-HA) in neurons show that the glial network volume size is restored to control volume in these animals. Quantification of the TM network volume is shown in panel J. Yellow arrows show glial network. Error bars show SD; ***$P < 0.0001$; **$P < 0.001$; and ns for nonsignificant. The data underlying this figure can be found in S1 Data. Genotypes: (A) w;; repo-Gal4, ihog-RFP/UAS-lacZ, (B) UAS-dEGFRλ, UAS-dp110CAAX;; repo-Gal4, UAS-ihog-RFP, (C) UAS-dEGFRλ, UAS-dp110CAAX; repo-Gal4/UAS-wg-RNAi, (D) UAS-armS10/ UAS-dEGFRλ, UAS-dp110CAAX; UAS-Fz1-RNAi; repo-Gal4, UAS-ihog-RFP.UAS-ihog-RFP, (G) w; >wg>wgNRT-HA, PaxRFP/ elav-lexA, lexAop-CD8-GFP; repo-Gal4, UAS-ihog-RFP/lexAop-flp, (H) UAS-dEGFRλ, UAS-dp110CAAX; >wg>wgNRT-HA, PaxRFP; repo-Gal4, UAS-ihog-RFP/ lexAop-flp, (I) UAS-dEGFRλ, UAS-dp110CAAX; >wg>wgNRT-HA, PaxRFP/ elav-lexA, lexAop-CD8-GFP; repo-Gal4, UAS-ihog-RFP/lexAop-flp. ELAV-GFP, embryonic lethal abnormal vision green fluorescent protein; Fz1, Frizzled1; TM, tumor microtube; Wg, wingless.

significant reduction in the number of synapses in adult (S7O Fig) and in the larvae under glioma conditions (Fig 5A, 5B and 5F). This effect is prevented by *Fz1* or *igl* down-regulation (Fig 5C, 5D and 5F).

Wg signaling is essential for synapse development and loss of Wg leads to dramatic reductions in target-dependent synapse formation [15], reviewed in the work by Libro and colleagues [17]. To confirm that glioma causes neurodegeneration through neuronal-Wg depletion via Fz1, we generated glioma brains while knocking down Fz1 (*UAS-Fz1-RNAi*),

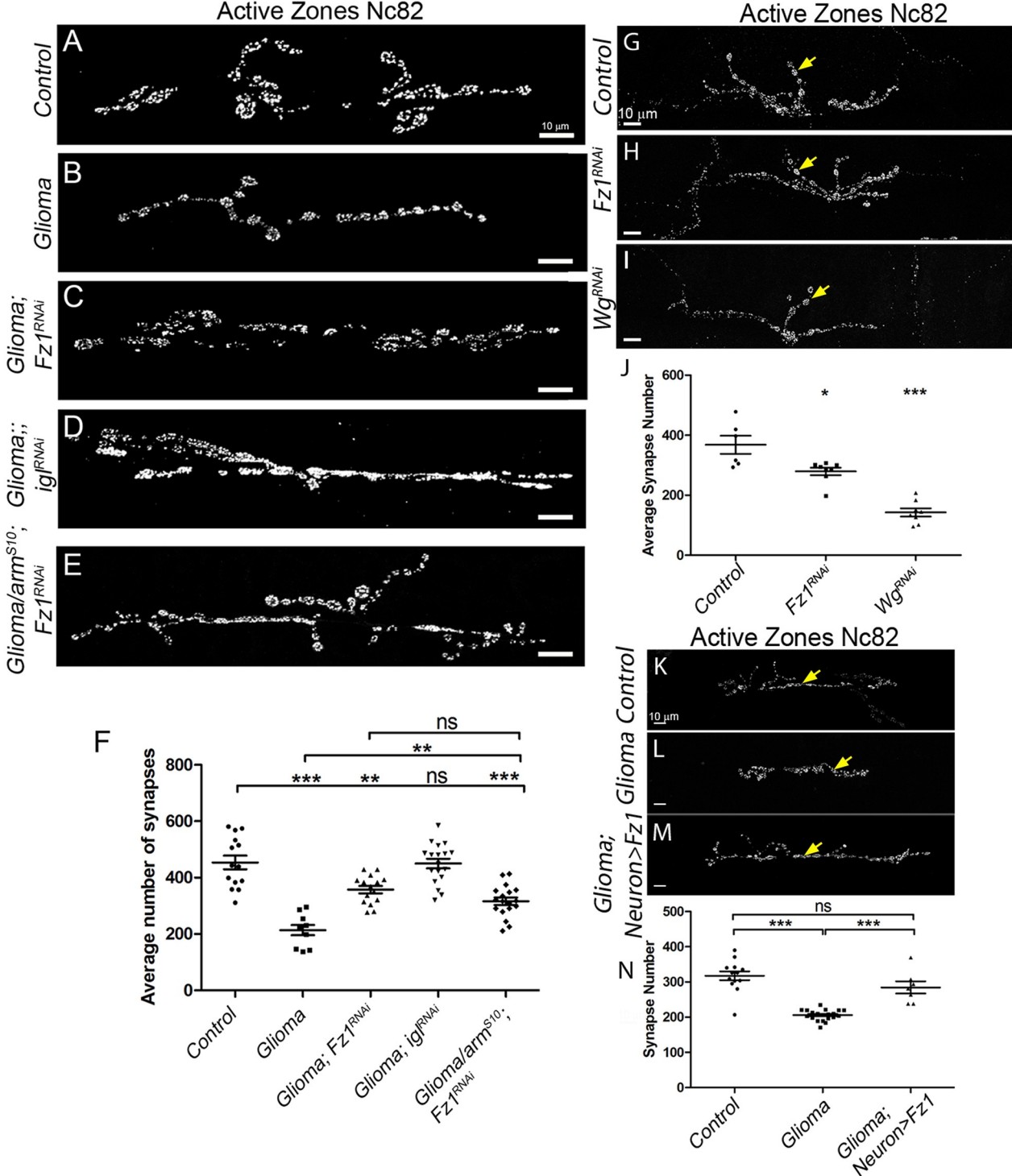

**Fig 5. Gliomas cause neurodegeneration and restoration of the glia-neuron Wg/Fz1 signaling equilibrium inhibits glioma progression and neurodegeneration.** Neurons from the larval neuromuscular junction are stained with Nc82 showing the synaptic active zones. (A–F) Upon glioma induction (B), the number of synapses (gray) is reduced when compared with the control (A). The number of synapses is restored upon knockdown in the glioma of Fz1 (C), igl (D), or armS10; Fz1-RNAi (E). The quantification of synapse number in all genotypes is shown in panel F. (G–J) Upon knockdown of wg (I) or Fz1 in D42 neurons (H), the number of synapses (gray) is reduced when compared with the control (G). Arrows indicate synapses. The quantification of synapse number in all genotypes is shown in panel J. (K–N) Upon glioma induction the number of synapses (gray) is reduced (L) when compared with the control (K). The number of synapses is restored upon overexpression of Fz1, specifically in the neurons (M). The quantification of synapse number is shown in panel N. Yellow arrows show active zones. Error bars show SD; ***$P < 0.0001$, **$P < 0.001$, *$P < 0.01$,

and ns for nonsignificant. The data underlying this figure can be found in S1 Data. Genotypes: (A) w;; repo-Gal4, ihog-RFP/UAS-lacZ, (B) UAS-dEGFRλ, UAS-dp110CAAX;; repo-Gal4, UAS-ihog-RFP, (C) UAS-dEGFRλ, UAS-dp110CAAX; UAS-Fz1-RNAi; repo-Gal4, UAS-ihog-RFP, (D) UAS-dEGFRλ, UAS-dp110CAAX;; repo-Gal4, UAS-ihog-RFP /UAS-igl-RNAi, (E) UAS-armS10/UAS-dEGFRλ, UAS-dp110CAAX; UAS-Fz1-RNAi; repo-Gal4, UAS-ihog-RFP.UAS-ihog-RFP, (G) w; UAS-CD8-GFP; D42-Gal4/UAS-lacZ, (H) w; UAS-CD8-GFP/Fz1-RNAi; D42-Gal4, (I) w; UAS-CD8-GFP/wg-RNAi; D42-Gal4, (K) w; repo-Gal4, ihog-RFP/UAS-lacZ, (L) UAS-dEGFRλ, UAS-dp110CAAX; lexAop-Fz1; repo-Gal4, UAS-ihog-RFP, (M) UAS-dEGFRλ, UAS-dp110CAAX; lexAop-fz1/ elav-lexA, lexAop-CD8-GFP; repo-Gal4, UAS-ihog-RFP. Fz1, Frizzled1; igl, igloo; Wg, wingless.

and, in addition, we activated the Wg pathway downstream by co-expressing *arm*$^{S10}$. The results show that the reduction of synapse number in glioma is prevented in glioma + *Fz1RNAi* + *arm*$^{S10}$ and it is similar to glioma + *Fz1RNAi* (Fig 5B, 5C, 5E and 5F). This result demonstrates that synapse decrease is specifically mediated by Fz1 accumulation and not to Wg signaling activation in glioma cells.

## Wg/Fz1 pathway disruption causes neurodegeneration

Neuronal development and physiology are dependent on Wg/Fz1 signaling, and disruptions in this signaling pathway lead to synapse loss, an early symptom of neurodegeneration (reviewed in the work by Libro and colleagues, the work by Kahn, the work by Arrazola and colleagues, and the work by Garcia and Arias [17,74–76]). To determine whether an imbalance in Wg distribution caused by glioma cells can affect the neighboring neurons, we determined the contribution of Fz1/Wg pathway in neural cell function. To inhibit Wg/Fz1 signaling, we expressed *UAS-fz1-RNAi* or *UAS-wg-RNAi* in motor neurons under the control of a *D42-Gal4* driver [46] and quantified the number of active zones (synapses) in the NMJ. The results showed a significant reduction in the number of synapses (Fig 5G–5J). These data show that Wg/Fz1 signaling in neurons is necessary for synaptogenesis.

Because the glioma cells vampirize Wg from the neural cells and the neural cells require Fz1-Wg signaling for synaptogenesis, we attempted to restore this signaling equilibrium by overexpressing Fz1 receptor in neurons surrounded by glioma cells. To avoid crossed expression, we generated *LexAop-Fz1* transgenic flies based on the LexA-LexAop genetic driver system [72] to target neurons. Fz1 was ectopically overexpressed specifically in neurons by using the *ELAV-LexA* driver, whereas the Gal4-UAS system was used to generate the glioma. Oversized glioma brains showed the expected glioma network compartmentalizing nests of neurons in the brain (S8A and S8B Fig). However, Fz1 overexpression in neurons restored the homogeneous distribution of Fz1 protein in the brain (S8C and S8D Fig) and prevented brain oversize (S8C and S8D Fig) and neuron nests (S8A'–S8D' Fig). In addition, Wg/Fz1 signaling equilibrium restoration partially rescued lethality, and most animals reached adulthood. To verify Fz1 activation of the pathway, we stained for Wg and Cyt-Arm (S8E–S8H Fig). As previously shown, glioma brains showed a heterogeneous distribution for Wg protein (S8E Fig), and, as a consequence, an imbalance in pathway activation reported by Cyt-Arm accumulation (S8G Fig). As expected, neuronal Fz1 overexpression in glioma brains restored Wg distribution and Cyt-Arm signal toward that of control brains (S8F and S8H Fig).

To further determine the effect of Wg/Fz1 signal restoration in neurons, we quantified the number of synapses in the NMJs. Synapse number reduction by the glioma is prevented by Fz1 overexpression in neurons (Fig 5K–5N). Altogether, these results indicate that the Wg/Fz1 pathway disruption caused by glioma is responsible for the synapse loss, and restoration of the signaling equilibrium between glia and neurons prevents synapse loss and therefore neurodegeneration.

## JNK activation in glioma

The mechanisms mediating TMs formation and progression are not yet well understood. However, the JNK signaling pathway is a potential major player in these processes. The JNK pathway is up-regulated in a number of tumors including GB, and it has been related to glioma malignancy [77–81]. Moreover, JNK is a target for specific drugs in combination with temozolomide treatments because it was proven to play a central role in GB progression [29,82–84]. However, little is known about the molecular mechanisms underlying JNK activation in glioma cells and the functional consequences for GB progression. Thus, we explored the contribution of JNK pathway activity to TMs infiltration through the brain and the mechanism of JNK activation.

To confirm JNK pathway activation in GB cells, we used the *Tre-RFP* reporter, which undergoes transcriptional activation in response to JNK signaling [85–87]. GB cells showed an up-regulation of Tre-RFP reporter (Fig 6A, 6B and 6F), indicating that JNK is activated in GB cells. Moreover, to determine whether JNK activation depends on TM formation or Wg pathway activity, we silenced *igl* or *Fz1*, respectively, and monitored *Tre-RFP* in GB brains (Fig 6C–6F). The results revealed that the increase in JNK activity depends on the formation of the TM network as well as the presence of Fz1 receptor in GB cells.

## The JNK pathway receptor Grindelwald mediates GB progression

To decipher the molecular mechanisms behind JNK activation in GB cells, we determined the contribution of the JNK receptor Grindelwald (Grnd) [88] to GB progression. We attenuated *grnd* expression with a *UAS-grnd-RNAi* expressed in glial cells under the control of *repo-Gal4*. The results showed that *grnd* knockdown prevented glial overgrowth upon GB induction (Fig 6G–6J and 6M). Additionally, in a *grnd* mutant background, glial cell proliferation was prevented (Fig 6K, 6L' and 6M). To further validate the contribution of JNK pathway in GB, we blocked JNK pathway signaling by expressing a dominant negative form of the effector Bsk (Bsk$^{DN}$) [89] in GB cells, by inducing GB in a background mutant for *eiger* (*egr*), which encodes the activating ligand of the JNK pathway, or by expressing the extracellular domain of Grnd (*grnd-extra*) [88], which causes a dominant negative effect, in GB cells. In all 3 situations, JNK pathway disruption in GB cells prevents the increase of glial cells number (S9 Fig). Further, we analyzed the contribution of *grnd* to Fz1 re-localization and the activity of Wg pathway (Cyt-armadillo localization) upon *grnd* depletion in GB (Fig 7). Fz1 is homogeneously distributed through the brain under normal conditions (Fig 7A and 7F), but, under GB conditions, Fz1 accumulates in the TMs surrounding the neurons (Fig 7B and 7F). *grnd* knockdown in GB cells or a *grnd−/−* background prevents Fz1 accumulation (Fig 7C, 7D and 7F). In addition, we blocked the JNK pathway downstream of grnd with Bsk$^{DN}$ [90] in GB cells, which also prevented Fz1 accumulation (Fig 7E).

Next, we addressed the JNK-Wg crosstalk in GB cells by analyzing the activity of Wg pathway, the re-localization of Fz1, and the formation of the TM network. Control brains exhibit a homogeneous signal for Cyt-Arm (Fig 7G, quantified in panel L, expressed as neuron/glia Cyt-Arm ratio), but, as a consequence of Fz1 re-localization, GB cells activate Wg pathway signaling, as indicated by the reduction of Cyt-Arm signal in GB cells and the increase of Cyt-Arm in neurons (Fig 7H, quantified in Fig 7L). The disruption of the JNK pathway by *grnd RNAi*, the *grnd−/−* mutant background or expression of *bsk$^{DN}$* prevented the increase of Wg signaling (neuron/glia Cyt-Arm ratio) in GB cells and decrease in neurons (Fig 7I–7L), consistent with the suppression of the Fz1 relocalization described above.

Furthermore, we analyzed the contribution of JNK pathway to the progression of TMs in GB. We used the reporter ihog-RFP to visualize cytonemes in WT glial cells and TMs in GB.

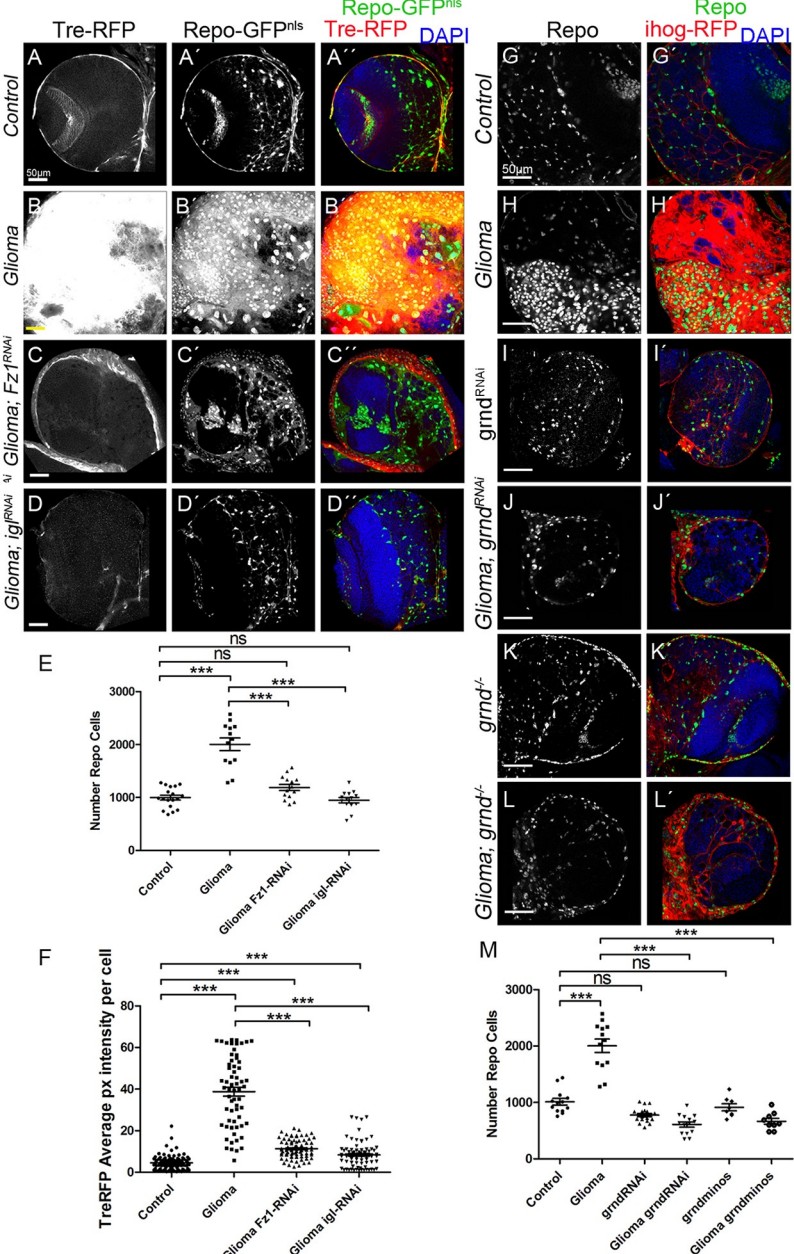

**Fig 6. JNK signaling pathway is activated in glioma via the TNF receptor Grindelwald which mediates GB progression.** Glial cell nuclei are labeled with UAS-GFP$^{NLS}$ (gray or green in the merge) driven by repo-Gal4. (A–D) JNK signaling pathway reporter TRE-RFP (gray or red in the merge) in control, glioma, glioma Gap43-RNAi, and Glioma Fz1-RNAi brain sections. The number of glial cells is quantified in panel E, and the TRE-RFP average pixel intensity per glial cells is quantified in panel F. (G–L) Glia are labeled with UAS-Ihog-RFP (red) driven by repo-Gal4 to visualize active cytonemes/TM structures in glial cells and stained with Repo (gray or green in the merge) in the following genotypes: control, glioma, grnd-RNAi, glioma grnd-RNAi, grnd−/− and Glioma grnd−/− brain sections. The number of Repo+ cells is quantified in panel M. Nuclei are marked with DAPI. Error bars show SD; $^{*}P < 0.01$, $^{**}P < 0.001$, $^{***}P < 0.0001$, or ns for nonsignificant. Scale bar size is indicated in this and all figures. The data underlying this figure can be found in S1 Data. Genotypes: (A) TRE-RFP; repo-Gal4, UAS-GFPNLS/UAS-lacZ, (B) UAS-dEGFRλ, UAS-dp110CAAX; TRE-RFP; repo-Gal4, UAS-GFPNLS, (C) UAS-dEGFRλ, UAS-dp110CAAX; TRE-RFP; repo-Gal4, UAS-GFPNLS/UAS-Gap43-RNAi, (D) UAS-dEGFRλ, UAS-dp110CAAX; TRE-RFP/ UAS-Fz1-RNAi; repo-Gal4, UAS-GFPNLS, (G) repo-Gal4, ihog-RFP/UAS-lacZ, (H) UAS-dEGFRλ, UAS-dp110CAAX;; repo-Gal4, UAS-ihog-RFP, (I) UAS-grnd-RNAi; repo-Gal4, ihog-RFP, (J) UAS-dEGFRλ, UAS-dp110CAAX; UAS-grnd-RNAi; repo-Gal4, UAS-ihog-RFP, (K) grndMINOS/grndMINOS; repo-Gal4, ihog-RFP, (L) UAS-dEGFRλ, UAS-dp110CAAX; grndMINOS/grndMINOS; repo-Gal4, UAS-ihog-RFP. Fz1, Frizzled1; GB,

glioblastoma; GFP, green fluorescent protein; grnd, Grindelwald; Ihog, interference hedgehog; JNK, cJun N-terminal kinase; TM, tumor microtube; TRE-RFP, RFP fluorescent protein regulated by a transcriptional response element of JNK pathway.

Upon down-regulation of *grnd* expression or the inhibition of JNK pathway by *Bsk*<sup>DN</sup> in GB cells, the formation of TMs was prevented (see ihog-RFP in Fig 7). These results suggest that JNK pathway activation mediated by the receptor Grnd is a requirement for TM formation. As a consequence of the prevention of TM network development, GB cells did not localize Fz1 in the areas of contact with neurons and were unable to mediate Wg depletion. Thus, Grnd receptor-JNK signaling is necessary in GB cells for tumor progression.

## MMPs are up-regulated in glioma

Next we explored the possibility that TM infiltration through the brain ECM could be mediated by the activity of a JNK signaling target, the MMPs [91]. To determine whether MMPs are expressed in GB, we used specific antibodies against *Drosophila* MMP1 or MMP2. The results revealed an increase in the glia/neuron ratio of MMP1 (Fig 8A–8D) and MMP2 (S10A, S10B, S10E and S10F Fig) in GB cells compared with control glial cells. MMPs colocalized with TMs and preferentially accumulated in the limiting region of GB and healthy brain tissue (Fig 8B and S10B Fig). To determine the hierarchy of events between TM formation and MMP expression, we attenuated *igl* expression in GB cells to prevent TM formation, which revealed that MMP1 and MMP2 protein levels are restored (Fig 8C and 8D and S10C and S10E Fig).

Because we have shown that GB cells vampirize Wg from neighboring neurons via Fz1 accumulation in TMs, to determine Wg pathway contribution to MMP1/2 expression, we silenced *Fz1* in GB and stained for MMPs. The results showed that MMP1 and MMP2 are not up-regulated upon expression of *Fz1 RNAi* (Fig 8C and 8D and S10D and S10E Fig). Next, we explored whether the accumulation of MMPs in glioma was due to a transcriptional up-regulation of MMP1 and MMP2. To determine this, we used the transcriptional reporters *MMP1-lacZ* and *MMP2-lacZ* in control and glioma, and the results show only a few cells with MMP1 and MMP2 transcriptional reporters activated in glial cells (arrowheads) in both control and glioma brains (S10G–S10J Fig). To corroborate these results, we undertook qPCRs experiments with *RNA* extracted from control and glioma larvae, which revealed no change in the transcription (*mRNA* levels) of *MMP1* or *MMP2* in glioma samples relative to the control (Fig 8E). Thus, MMP proteins are up-regulated post-transcriptionally.

To determine whether Fz1 is sufficient to trigger MMP up-regulation, we overexpressed Fz1 in normal glial cells (Fig 8F and 8G). The results revealed MMP1 and MMP2 increase in glial cells upon Fz1 overexpression compared with control samples (Fig 8F–8H and S10K–S10M Fig). Altogether, the results suggest that MMP1 and MMP2 protein accumulation in glial cells depends on TM network formation and Fz1-mediated Wg signaling.

## JNK triggers MMPs expression in GB cells to expand the TM network

Glioma TMs cross the ECM and infiltrate territories distant from the original GB site. Inspired by the identification of MMP1 as a target of JNK signaling in epithelial cells [32], we considered whether that could be the case in GB cells. We hypothesized that JNK activity could mediate the production of MMPs to facilitate ECM digestion and thus allow TMs infiltration through the brain.

To validate this hypothesis, we determined whether MMPs expression in GB depends on Grnd and JNK pathway signaling. We blocked the JNK pathway with the expression of *bsk*<sup>DN</sup> or *grnd RNAi* in GB cells (Fig 9C and 9E compared with controls Fig 9A, 9B and 9D). The

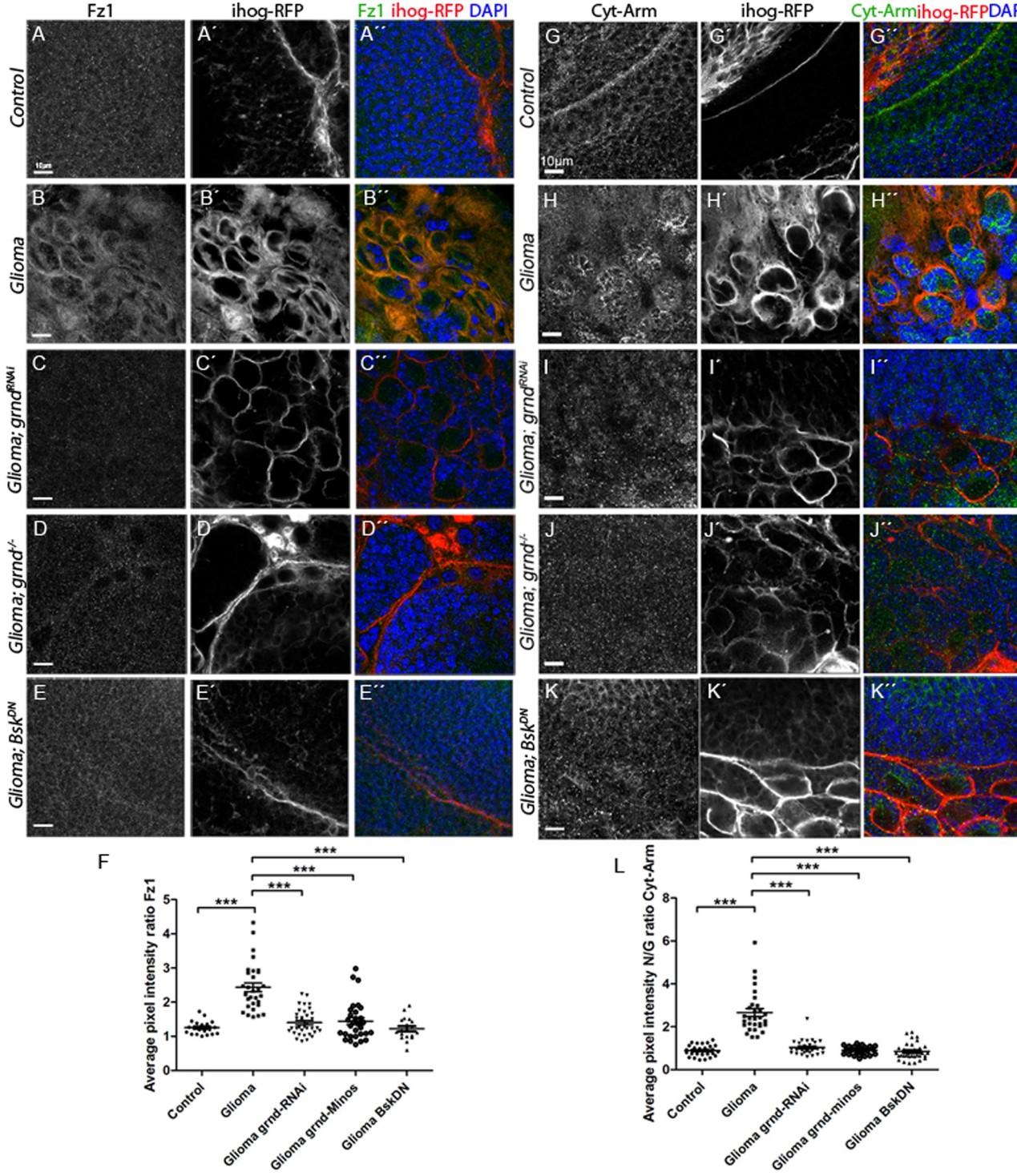

**Fig 7. JNK pathway is necessary to localize Fz1 in the TMs, grnd mediates Wg pathway activation in GB and promote GB progression.** Brains from third instar larvae displayed at the same scale. Glia are labeled with UAS-Ihog-RFP (gray or red in the merge) driven by repo-Gal4 to visualize active cytonemes/TM structures in glial cells and stained with Fz1 (gray or green in the merge, panels A–F) and Cyt-Arm (gray or green in the merge, panels G–K) in the following genotypes control, glioma, glioma grnd-RNAi, Glioma grnd−/− and Glioma Bsk^DN brain sections. Fz1 and Cyt-Arm average pixel intensity staining quantification ratio between ihog+ and ihog–domains (F and L). (A, G) Fz1 and Cyt-Arm is homogeneously distributed in control sections. (B, H) In glioma brains, Fz1 accumulates in the TMs, and Cyt-Arm accumulates in the neurons' cytoplasm where it is inactive. (C–D, I–J) Knockdown or knockout of grnd and Bsk^DN in glioma brains showing a normal glial network, and Fz1 and Cyt-Arm does not accumulate showing a homogeneous distribution similar to the control. Fz1 average pixel intensity quantification for glia/neuron ratio (F) and Cyt-Arm average

pixel intensity quantification for neuron/glia ratio between ihog- and ihog+ domains (L). Nuclei are marked with DAPI (blue). Error bars show SD; *$P$ < 0.01, **$P$ < 0.001, ***$P$ < 0.0001, or ns for nonsignificant. Scale bar size is indicated in this and all figures. The data underlying this figure can be found in S1 Data. Genotypes: (A, G) repo-Gal4, ihog-RFP/UAS-lacZ, (B, H) UAS-dEGFRλ, UAS-dp110CAAX;; repo-Gal4, UAS-ihog-RFP, (C, I) UAS-dEGFRλ, UAS-dp110CAAX; UAS-grnd-RNAi; repo-Gal4, UAS-ihog-RFP, (D, J) UAS-dEGFRλ, UAS-dp110CAAX; grndMINOS/grndMINOS; repo-Gal4, UAS-ihog-RFP, (E, K) UAS-dEGFRλ, UAS-dp110CAAX;; repo-Gal4, UAS-ihog-RFP/ UAS-bskDN. Bsk$^{DN}$, dominant negative form of the effector Bsk; Cyt-Arm, Cytoplasmic-Armadillo; Fz1, Frizzled1; GB, glioblastoma; grnd, Grindelwalkd; Ihog, interference hedgehog; JNK, cJun N-terminal kinase; RFP, Red Fluorescent Protein; TM, tumor microtube; Wg, wingless.

quantification showed that MMP1 up-regulation is prevented upon JNK silencing (Fig 9F). Thus, as in epithelial cells, MMP1 is a target of the JNK pathway in GB cells.

To determine the contribution of MMPs to Fz1 receptor accumulation and Wg vampirization, we silenced *MMP1* or *MMP2* in GB and stained for Fz1 or Wg (Fig 10A–10G and S11A–S11E Fig). We also analyzed the volume of the TM network (see ihog-RFP in Fig 10A–10F and 10H), and the data showed that *MMP1* or *MMP2* knockdown prevents Fz1 accumulation in GB cells, TM network formation, and expansion in GB (Fig 10A–10H).

Moreover, Wg accumulation in GB membranes required *MMP1* or *MMP2* (S11A–S11E Fig). As expected, *MMP1 or MMP2 RNAi* also prevent the imbalance of Wg signaling (S11F–S11J Fig), similar to previous results obtained for Fz1 distribution.

To determine the contribution of MMPs to the lethality caused by GB, we down-regulated *MMP1* or *MMP2* and analyzed survival of adult flies. To avoid developmental defects caused by *MMP1* or *MMP2* attenuation, we induced the GB condition at day 4 of adulthood using the *Gal80$^{TS}$* repressor (see Methods and materials). Glioma lethality was rescued by *MMP1* or *MMP2* down regulation (Fig 10I). Finally, we addressed the role of MMPs in neurodegeneration. We specifically inhibited *MMP1* or *MMP2* in GB and quantified the number of active zones in NMJ. The results showed that the number of active zones is rescued compared to GB (Fig 10J and S11K–S11P Fig).

Taking all these data together, we have shown that MMPs mediate GB progression and neurodegeneration. Further, *MMP* silencing in GB cells is sufficient to rescue life span in adult flies. Altogether, these results reveal a novel positive feedback loop involving Wg, JNK, and MMPs that contribute to GB development and progression in the fly model. These discoveries reveal potential novel mechanisms underlying GB progression.

## Discussion

In addressing the mechanism by which GB cells infiltrate into the brain and affect the neighboring neurons, we have shown in the *Drosophila* glioma model that GB cells display a network of TMs that enwrap neurons, accumulate Fz1, and vampirize neuronal Wg causing neurodegeneration, as evidenced by TEM images, Wg and Fz1 staining, Wg pathway activity reporters, GRASP, and PLA experiments. Furthermore, we showed that reducing Wg in *Drosophila* glioma cells is dispensable but reducing Fz1 prevents tumor progression. However, reducing Wg or Fz1 in neurons results in loss of synapses, indicating that the Wg/Fz1 pathway disruption caused by glioma is responsible for the synapse loss, and restoration of the signaling equilibrium between glia and neurons prevents neurodegeneration. Moreover, in addressing the mechanism by which GB cells infiltrate into the brain, we have shown that GB cells establish a positive feedback loop to promote their TMs expansion, in which the Wg pathway activates JNK in GB cells, and then JNK signaling leads to the post-transcriptional up-regulation and accumulation of MMPs, which facilitate TMs infiltration throughout the brain allowing further Wg depletion from neurons. These conclusions are evidenced by the JNK reporter, RFP fluorescent protein regulated by a transcriptional response element of JNK pathway (TRE-RFP), MMP staining, and by functional experiments showing that reducing JNK/MMPs

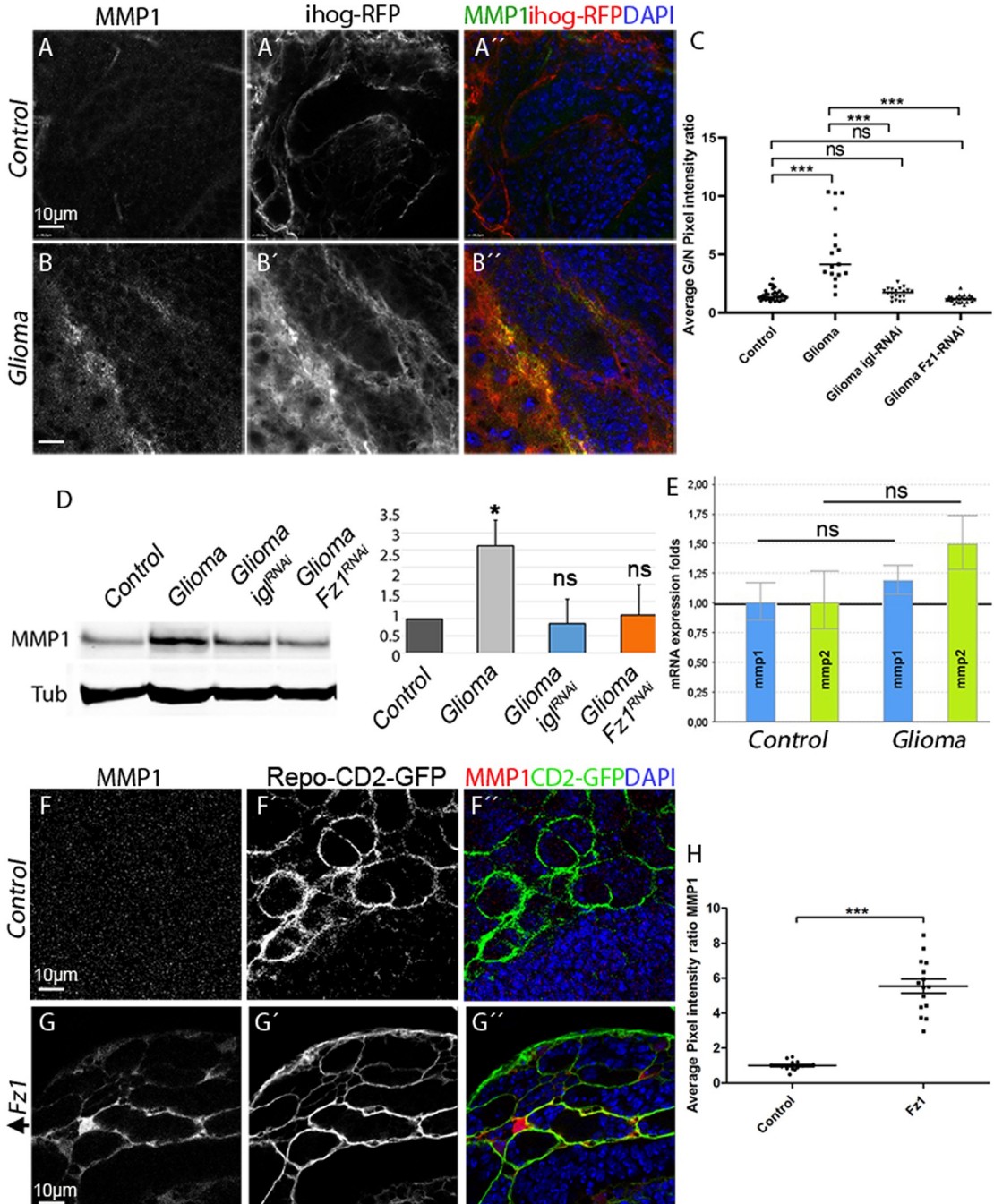

**Fig 8. MMP1 is up-regulated in GB, and the Wg pathway is necessary and sufficient to activate MMP expression in glial cells.**
Brains from third instar larvae displayed at the same scale. (A) Glia are labeled with UAS-Ihog-RFP (gray or red in the merge) driven by repo-Gal4 to visualize active cytonemes/TM structures in glial cells and stained with MMP1 (gray or green in the merge) is homogeneously distributed in control sections, with a slight accumulation in the Ihog+ projections. (B) MMP1 accumulates in the TMs and specifically in the projections that are in contact with the neuronal clusters. (C) Quantification of MMP1 staining ratio between ihog+ and ihog−domains. (D) Western blot of samples extracted from control, glioma, glioma Gap43-RNAi and glioma Fz1-RNAi larvae showing changes in the amount of MMP1 or protein. (E) qPCRs with RNA extracted from control and glioma larvae showing no change in the transcription (mRNA levels) of MMP1 or MMP2. (F–G) Glial cell bodies and membranes labeled with CD8-GFP in gray (green in the merge) driven by repo-Gal4 to the glial cells and stained with MMP1 (gray or red in the merge). (F) MMP1 staining in control brain. (G) MMP1 accumulates in the glial cells upon Fz1 overexpression. (H) Quantification of MMP1 staining ratio between GFP+ and GFP−domains. Nuclei are marked with DAPI (blue). Error bars show SD; $^{*}P < 0.01$, $^{**}P < 0.001$, $^{***}P < 0.0001$, or ns for nonsignificant. Scale bar size is indicated in this and all figures. The data underlying this figure

can be found in S1 Data. Genotypes: (A-B) repo-Gal4, ihog-RFP/UAS-lacZ, (A-C) UAS-dEGFRλ, UAS-dp110CAAX;; repo-Gal4, UAS-ihog-RFP, (A) UAS-dEGFRλ, UAS-dp110CAAX;; repo-Gal4, UAS-ihog-RFP /UAS-Gap43-RNAi, (A) UAS-dEGFRλ, UAS-dp110CAAX; /UAS-Fz1-RNAi; repo-Gal4, UAS-ihog-RFP, (F) w;; Repo-LexA, LexAop-CD2-GFP, (G) w;; Repo-LexA, LexAop-CD2-GFP/LexAop-Fz1. Fz1, Frizzled1; GB, glioblastoma; GFP, green fluorescent protein; Ihog, interference hedgehog; MMP, matrix metalloproteinase; qPCR, quantitative Polimerase Chain Reaction; RFP, Red Fluorescent Protein; TM, tumor microtube; Wg, wingless.

activity suppresses tumor growth and TM network expansion. Our findings in *Drosophila* reveal a molecular mechanism for TMs infiltration, which explains both neuron-dependent tumor progression and the neurodegeneration associated with GB.

If confirmed in humans, TM formation and the cellular-signaling consequences could serve as potential targets in the future to treat GB patients. Therapies targeting receptor tyrosine kinase (RTK) signaling are effective in treating breast and lung cancer [92,93]. However, the benefit from these strategies is limited because of resistance mechanisms that allow tumoral cells to survive and expand [94,95]. Thus, a deeper knowledge on the specific signals activated by RTK for each particular tumor progression is a need.

Sustained JNK pathway signaling is required for EGFR to bypass resistance [96], which suggest JNK as a candidate for anticancer therapies. JNK and MMPs are activated by several pathways, and a crosstalk action of these pathways in humans could contribute to GB progression. In consequence, it is important to determine whether TM and WNT signaling are the main activators of JNK and MMPs in human GB cells to determine the relevance of this feedback loop in patients.

Consistent with our findings presented here on the vampirization of Wg from the neurons by glia, recent evidence indicates that microenvironment signals contribute to glioma progression. One of these signals is Neuroligin-3 (NLGN3), a synaptic protein that is cleaved and secreted after neuronal activity and promotes PI3K- Target of Rapamycin (TOR) signaling, stimulating glioma growth. Thus, NLGN3 mediates an autocrine/paracrine loop in glioma cells that perpetuates tumoral features [97] (reviewed in the work by Johung and Monje [43]). Also, neural precursor cells (NPCs) from the subventricular zone (SVZ) produce chemoattractants (Secreted Protein Acidic And Cysteine Rich [SPARC/SPARCL1], Heat Shock Protein 90B [HSP90B], and pleiotrophin) that facilitate glioma invasion of the SVZ through rhomboid/Rho associated kinase (Rho/ROCK) signaling [98]. Also, *Tweety homologue-1* (*TTHY1*) expression in GB cells, potentially mediated by NLGN3 [38], regulates TM formation [99].

Our study shows that TMs intercalate among neurons and enwrap them in perineuronal nests, thereby establishing an intimate glioma-neuron linkage. We show that GB cells make direct contact via TMs and deprive (vampirize) neurons of WNT. Moreover, JNK pathway activation, through the receptor Grnd, and the expression of the JNK targets, MMP1 and MMP2, in GB cells are required for TMs network formation and infiltration. As a consequence, Wg vampirization and Wg pathway activation respond to JNK and TM expansion, and these 3 events constitute a regulatory positive feedback loop in GB progression (scheme in Fig 10K), which is initiated by the founder mutations (*EGFR$^λ$/dp110$^{CAAX}$*), already known to be involved in F-Actin polymerization because the downstream effector of the pathway, Akt, phosphorylates many proteins involved in polymerization and stabilization of the Actin cytoskeleton [100–102].

Expression data from human cancer databases indicate that glioma cells do not up-regulate canonical WNT pathway ligands, neither up-regulate its receptors' expression; however, downstream targets of the pathway such as β-catenin are activated, indicating that WNT canonical pathway is active. Recent publications address the contribution of noncanonical WNT pathway components to GB progression and prognosis [103] and the expression of

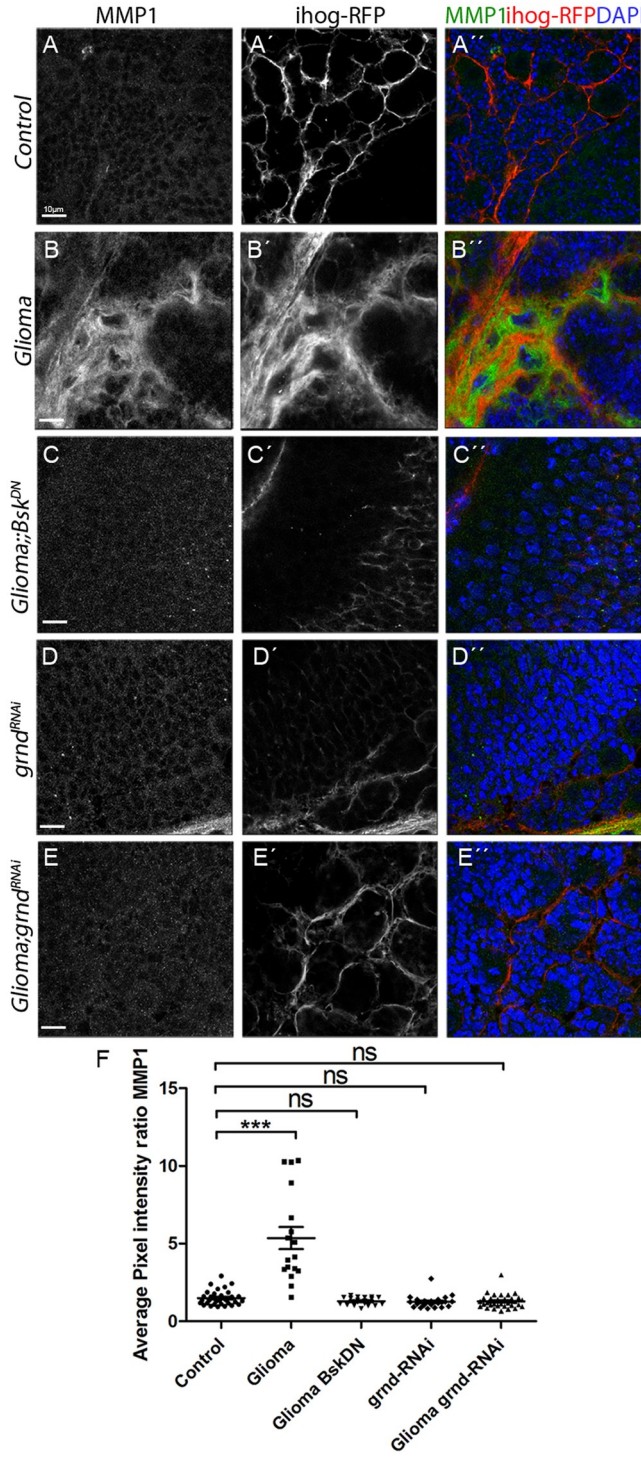

**Fig 9. JNK triggers MMPs expression in GB.** Brains from third instar larvae displayed at the same scale. Glia is labeled with UAS-Ihog-RFP (gray or red in the merge) driven by repo-Gal4 to visualize active cytonemes/TM structures in glial cells and stained with MMP1 (gray or green in the merge). (A) MMP1 is homogeneously distributed in control sections. (B) MMP1 accumulates in the TMs and specifically in the projections that are in contact with the neuronal clusters. (C–E) Blocking JNK pathway by using a UAS-bsk[DN] or UAS-grnd-RNAi in glioma brains restores a normal glial network and MMP1 does not accumulate showing a homogeneous staining along the brain section. (F) Quantification of MMP1 staining ratio between ihog+ and ihog–domains. Nuclei are marked with DAPI (blue). Error bars show SD; $^*P < 0.01$, $^{**}P < 0.001$, $^{***}P < 0.0001$, or ns for nonsignificant. Scale bar size are indicated in this and

all figures. The data underlying this figure can be found in S1 Data. Genotypes: (A) repo-Gal4, ihog-RFP/UAS-lacZ, (B) UAS-dEGFRλ, UAS-dp110CAAX;; repo-Gal4, UAS-ihog-RFP, (C) UAS-dEGFRλ, UAS-dp110CAAX;; repo-Gal4, UAS-ihog-RFP/ UAS-bskDN, (D) UAS-grnd-RNAi; repo-Gal4, ihog-RFP, (E) UAS-dEGFRλ, UAS-dp110CAAX; UAS-grnd-RNAi; repo-Gal4, UAS-ihog-RFP. bsk^{DN}, dominant negative form of the effector Bsk; GB, glioblastoma; grnd, Grindelwald; Ihog, inference hedgehog; JNK, cJun N-terminal kinase; MMP, matrix metalloproteinase; repo-Gal4,; RFP, Red Fluorescent Protein; TM, tumor microtube.

WNT 2b and 5a ligands in GB cells. Instead, the results in the fly model show that glioma cells relocate Fz1 receptor in the TMs, allowing depleting Wg from neurons. Thus we use the term vampirization as the action of GB cells to exhaust or prey upon healthy cells (neurons) in the manner of a vampire, because they drain Wg/WNT and cause the demise of the neurons. Consistent with these data, in the patient-derived GB xenograft model, in which WNT1 is deposited in GB cells and the WNT pathway is activated, β-catenin is up-regulated. The available data suggest that GB TMs grow toward the source of Wg. However, as TMs expand upon Fz1/Wg signaling, the question regarding the exact order of events remains open. Do TMs require some initial stimuli from the source of Wg to grow? Alternatively, do TMs initiate growth triggered by glial internal signals and are directed through a gradient of neuron-secreted attractants? Do TMs only mediate WNT canonical pathway ligands depletion?

Concerning the mechanism of Wg vampirization, we have expressed a nonsecretable, HA-tagged version of membrane-tethered Wg^{NRT} in neurons. In this experiment, Wg is anchored to neuronal membranes demonstrating that GB cells take Wg directly from the neurons. However, further studies are still required to determine the precise mechanism of neuronal-Wg depletion by GB cells TMs.

Frizzled receptors mediate matrix metalloproteinases expression and modulate cell migration through the basement membrane including T-cell extravasation [104], neural stem cell proliferation and migration in hypoxia [105], and pulmonary fibrosis in lung epithelial cells [106]. MMP expression is associated with GB invasion, growth, and angiogenesis [107,108] and emerge as promising clinical targets in combination with treatment with the chemotherapy drug, temozolomide [109]. Furthermore, phase II clinical trials using the broad-spectrum MMP inhibitor marimastat, in conjunction with temozolomide has shown encouraging results [110]. The accumulation of *MMPs* is triggered by JNK-Wg-TMs and in turn facilitates the infiltration of TMs in *Drosophila* GB cells. *MMP1* or *MMP2* knockdown reduces the volume occupied by TMs and Wg signaling with the concomitant consequences: reduction of GB progression, prevention of neurodegeneration (synapse loss), and lethality. Consequently, MMPs are part of the positive feedback loop in GB and mediate the equilibrium among Wg signaling, JNK, and TM expansion. Thus, we have shown that JNK pathway activation, and MMP secretion in consequence, is required for TMs expansion and vice versa. The mechanisms that lead to the establishment of this loop could be various, including the combined activation of EGFR and PI3K pathways (Phosphatase and tensin homolog [PTEN] inhibition). We propose that GB cells become addicted to signals independent of the founder mutations (PI3K and EGFR), in this case, but dependent on the positive feedback loop formed by Wg pathway, JNK, MMPs, and the TMs (Fig 10K). As a consequence, treatments tackling EGFR or PI3K have limited success [98,99], because GB cells become reliant on Wg, JNK, and MMPs. This particular signaling loop is required for GB cells to progress, although it is dispensable for normal glia development. Thus, our discovery of the importance of Wg depletion via TMs and the requirement of JNK and MMPs in TMs formation and glioma progression now reveals TMs formation as a targetable feature for GB treatments.

It is widely observed that brain tumors and related ailments can cause cognitive decline and neuronal dysfunction (reviewed in the work by Bergo and colleagues [111]). High-grade

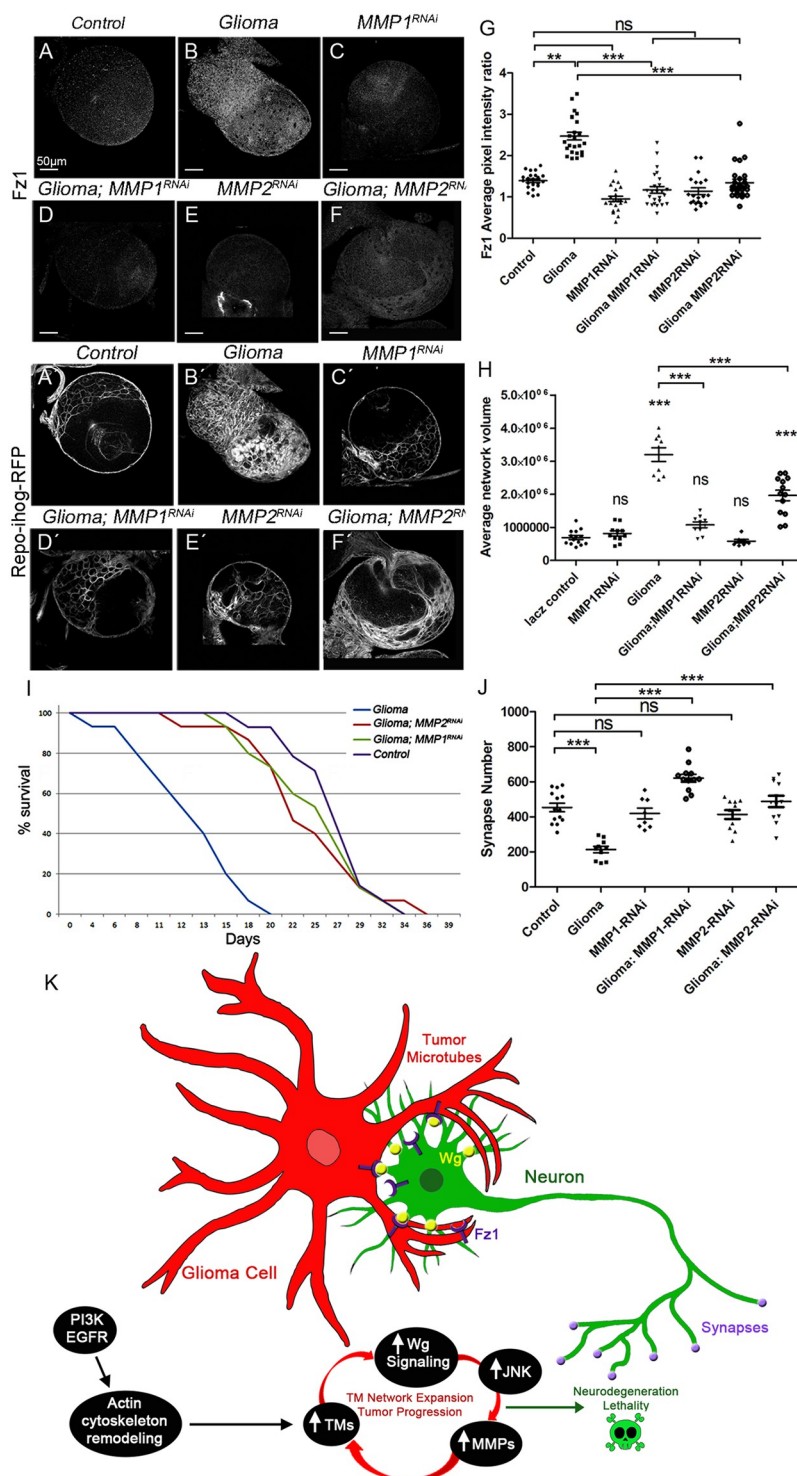

**Fig 10. TM expansion and synapse loss in the neurons requires MMPs in glioma.** Brains from third instar larvae displayed at the same scale. Glia are labeled with UAS-Ihog-RFP (gray) driven by repo-Gal4 to visualize active cytonemes/TM structures in glial cells and stained with Fz1 (gray) in the following genotypes (A–F) control, glioma, MMP1-RNAi, glioma MMP1-RNAi, MMP2-RNAi, and glioma MMP2-RNAi brain sections. (G) Quantification of Fz1 average pixel intensity staining ratio between ihog+ and ihog–domains. (H) Quantification of glial/glioma network volume expansion. (I) Survival curve of adult control, glioma, glioma MMP1-RNAi, and glioma MMP2-RNAi flies after a number of days of glioma induction and progression. (J) Neurons from the larval neuromuscular junction were stained with Nc82 (brp, in gray) to reveal and quantify the synaptic active zones. Upon glioma induction, the number

of synapses is reduced when compared with the control. The number of synapses is restored toward control levels upon knockdown of MMP1 or MMP2. (K) Model: Glioma cells produce a network of TMs that grow to reach neighboring neurons. Intimate membrane contact facilitates neuronal-Wg sequestering mediated by glioma Fz1 receptor. Glial cells are initially transformed into malignant GB upon EGFR and PI3K pathways constitutive activation; afterward GB cells establish a positive feedback loop including TMs, Wg signaling, JNK, and MMPs. Initial stimulation of actin cytoskeleton remodeling via EGFR/PI3K enables initial expansion of TMs; as a consequence, Fz1 receptor accumulation in TMs mediates neuronal-Wg depletion and Wg signaling up-regulation in the GB cells, which activates JNK in GB. As a consequence, MMPs are up-regulated and facilitate further TM infiltration in the brain; hence the GB TM network expands and mediate further wingless depletion to close the loop. Error bars show SD; $^*P < 0.01$, $^{**}P < 0.001$, $^{***}P < 0.0001$, or ns for nonsignificant. Scale bar size is indicated in this and all figures. The data underlying this figure can be found in S1 Data. Genotypes: (A) repo-Gal4, ihog-RFP/UAS-lacZ, (B) UAS-dEGFRλ, UAS-dp110CAAX;; repo-Gal4, UAS-ihog-RFP, (C) UAS-MMP1-RNAi; repo-Gal4, ihog-RFP, (D) UAS-dEGFRλ, UAS-dp110CAAX; UAS-MMP1-RNAi; repo-Gal4, UAS-ihog-RFP, (E) UAS-MMP2-RNAi; repo-Gal4, ihog-RFP, (F) UAS-dEGFRλ, UAS-dp110CAAX; UAS-MMP2-RNAi; repo-Gal4, UAS-ihog-RFP. EGFR, Epidermal Growth Factor Receptor; Fz1, Frizzled1; GB, glioblastoma; Ihog, interference hedgehog; JNK, cJun N-terminal kinase; MMP, matrix metalloproteinase; PI3K, phosphatidylinositol-3 kinase; RFP, Red Fluorescent Protein; TM, tumor microtube; Wg, wingless.

glioma patients continue to display cognitive deficits after surgery, radiotherapy, or chemotherapy [112–114]. The most common deficits concern memory, executive functions, and general attention beyond the effects of age, education, and gender [115].

Synapse loss is an early step in neurodegeneration [116–118] that is consistent with the cognitive defects observed in GB patients. Nonetheless, cognitive defects can be observed also in patients with excess of synapses as in the case of fragile X syndrome [119,120]. GB cells can stimulate aberrant synapses associated with seizures [121], which are consistent with cognitive dysfunctions. There is preclinical work [122,123] supporting glioma-induced neuronal death due to glutamate cytotoxicity, in addition clinical studies from Robert and colleagues [124] support neuronal death in GB patients. However, it is certainly very difficult to draw clear conclusions from clinical samples or clinical courses, considering that therapy, antiepileptics, and the pure space occupation plus the edema contribute to the neuronal dysfunction, degeneration, and cell death.

In particular, neuronal cell loss is typically found at and around glioblastomas [73,125], and neurocognitive disturbances are a frequent finding in glioma patients [126–128]. Although evidences from our experience and from neuropathology expertise support these observations, this is an open debate that requires further attention.

We showed recently that TM network formation determines GB tumor malignancy, confers radiotherapy resistance, and is associated with the known prognostic differences of glioma subgroups [7]. TM stability in GB is sensitive to *GAP43* expression in tumoral cells [7]. Our data also reveal that re-establishing Wg signaling equilibrium by Fz1 overexpression in neurons not only restores synapse number but also blocks GB progression. Functional disruption of the equilibrium between Wg signaling in GB glia and neurons is described here for the first time. Possibly, this mechanism could also affect other molecules related to tumor progression that signal through cytonemes, such as Notch, Hedgehog, or transforming growth factor beta (TGFβ). Moreover, cytoneme-like structures initially described in epithelial cells, also play a role in development and in the adult ovary, trachea, or myoblasts [129,130]. Hence, we propose that cytoneme-like structures in physiological conditions and TMs in pathological GB conditions could redistribute limited amounts of signaling molecules among competing cell types; therefore cytoneme-mediated long-range redistribution of signaling molecules could be a general mechanism for the cells to compete for various resources.

This study integrates for the first time the oncogenic nature of glioma with the neuronal degeneration caused by TM expansion. This innovative concept of glioma-induced neurodegeneration opens the possibility of combined treatments to fight GB progression and associated neurodegeneration at the same time. Our data demonstrate that making neurons more

competitive for secretable factors, such as Wg, has an impact in GB tumor growth, although whether this can be manipulated by small molecule treatments remains to be determined.

The rapid transformation of GB cells and the heterogeneity of mutations in these tumors are a handicap for targeted therapies. In our view, cellular features such as the network shared by GB cells emerge as an alternative to tackle tumor progression. Among the possible new strategies, TM dynamic and cellular transport of receptors to the TMs could be a target to prevent GB proliferation and neurodegeneration. GAP43 has emerged as a functional component of GB TM network formation; this may be a good protein to target with small-molecule inhibitors to reduce the TMs and therefore GB growth and neural degeneration. Likewise, recent studies indicate that other proteins such as Flotillin, a scaffolding protein within caveolar membranes and involved in the formation of caveolae or caveolae-like vesicles, can also participate in cytoneme dynamics [52,131].

In summary, Wg, JNK, and MMPs form a self-perpetuating loop that plays a central role in GB progression. TM network facilitates Wg signaling in GB and therefore the initiation of the loop. These complex glioma- and brain-specific microenvironmental interactions underscore the notion that this tumor type bears very specific biological features, with TMs exerting a central role. Therapies that specifically inhibit the vampirization of neuronal Wg by tumor cells, either by targeting of TMs or other mechanisms, emerge as a promising avenue to treat this difficult disease and also its side effects more effectively in the future.

## Materials and methods

### Fly stocks

Flies were raised in standard fly food at 25 ˚C.

Fly stocks from the Bloomington stock Centre: UAS-GFPnls (BL4776), UAS-lacZ (BL8529), UAS-myr-RFP (BL7119), UAS-igl-RNAi (BL29598), arm-GFP (BL8555), nkd04869a-lacZ (BL25111), D42-Gal4 (BL8816), GFP-fz1-GFP (BL59780), repo-Gal4 (BL7415), UAS-CD8-GFP (BL32186), tub-gal80ts (BL7019), elav-lexA (BL52676), lexAop-CD8-GFP (BL32205), lexAop-flp (BL-55819), UAS-armS10 (BL4782), sqh-GFP (BL57145), UAS-CD4-spGFP1-10, lexAop-CD4-spGFP11 (obtained from BL58755), lexAop-CD2-GFP (BL32205), UAS-bskDN (BL9311), egr[MI15372] (BL59754), MMP2-GFP (BL36162), MMP1-lacZ (BL12205), and MMP2-lacZ (BL10358). Fly stocks from the Vienna Drosophila Resource Centre: UAS-fz1-RNAi (v105493), fz4-GFP (v318152), UAS-wg-RNAi (v104579), UAS-yellow-RNAi (v106068), UAS-nrg-RNAi (v107991), UAS-mmp1-RNAi (v101505), UAS-mmp2-RNAi (v107888), lexAop-Fz1 (this study), GFP-sls (MLC, ZCL2144 from http://flytrap.med.yale.edu), UAS-dEGFRλ, UAS-PI3K92ECAAX (dp110CAAX) (A gift from R. Read), UAS-ihog-RFP and lifeactin-GFP (a gift from I. Guerrero), tsh-lacZ and dally-lacZ (gifts from M. Milan), FRT Wg FRT NRT–Wg-HA, pax–Cherry (a gift from A. Baena-López), and Repo-lexA (a gift from C Klambt). For electron microscopy studies, we used the UAS-HRP:CD2 as reporter [132], UAS-GPI-YFP [133], UAS-GMA-GFP [134], TRE-RFP-1b (a gift from J.P. Vincent), UAS-grnd-RNAi, grndMINOS, and UAS-grndEXTRA (a gift from P. Leopold) [88].

### *Drosophila* GB model

The most frequent genetic lesions in human gliomas include mutation or amplification of the EGFR gene. Glioma-associated EGFR mutant forms show constitutive kinase activity that chronically stimulates Ras signaling to drive cell proliferation and migration [135,136]. Other common genetic lesions include loss of the lipid phosphatase PTEN, which antagonizes the phosphatidylinositol-3 kinase (PI3K) signaling pathway, and mutations that activate PI3KCA, which encodes the p110a catalytic subunit of PI3K [135,136]. Gliomas often show

constitutively active Akt, a major PI3K effector. However, EGFR-Ras or PI3K mutations alone are not sufficient to transform glial cells. Instead, multiple mutations that co-activate EGFR-Ras and PI3K/Akt pathways are required to induce a glioma [48]. In *Drosophila*, a combination of EGFR and PI3K mutations effectively causes a glioma-like condition that shows features of human gliomas, including glia expansion, brain invasion, neuron dysfunction, synapse loss, and neurodegeneration [44,137,138]. Moreover, this model has proved to be useful in finding new kinase activities relevant to glioma progression [47]. To generate a glioma in *Drosophila melanogaster* adult flies, we used the Gal4/UAS system [45] as described above (repo-Gal4>UAS-EGFRλ, UAS-dp110). To restrict the expression of this genetic combination to adulthood, we used the thermo sensitive repression system Gal80TS. Individuals maintained at 17 ˚C did not activate the expression of the UAS constructs, but when flies were switched to 29 ˚C, the protein Gal80TS changed conformation and was no longer able to bind to Gal4 to prevent its interaction with UAS sequences, and the expression system was activated.

### Generation of transgenic flies

LexAop-Fz1 construct was generated by RECOMBINA SL Fz1 (CG17697) CDS was synthesized by overlapping g-block assembly. The complete 17665 bp fragment was amplified using the high fidelity Phusion taq polymerase (Thermo fisher Scientific) and Eco.Friz.Fw and Xba.Friz.Rv primers. PCR amplicon was cloned in pJET entry vector (Thermo Fisher Scientific), and then Frizzled fragment was released with EcoRI/XbaI restriction enzymes and subcloned into destination pLOTattB plasmid.

Eco.Fz1.Fw 5´-GAATTGGGAATTCATGTGGCGTCAAATCCTG-3´

Xba.Fz1.Rv 5´-TCTAGACTAGACGTACGCCTGCGCCC-3´

Transgenic flies were injected, and Fz1 fragment was inserted in Chromosome 2L by the *Drosophila* microinjection service (CBMSO-CSIC) using the following stock:

y[1] M[vas-int.Dm]ZH-2A w[*]; M[3xP3-RFP.attP']ZH-22A (BL24481). Transgenic flies were selected individually by eye color (w+) and balanced with CyO.

### Antibodies for immunofluorescence

Third-instar larval brains were dissected in phosphate-buffered saline (PBS), fixed in 4% formaldehyde for 30 min, washed in PBS + 0.1 or 0.3% Triton X-100 (PBT), and blocked in PBT + 5% BSA.

Antibodies used were: mouse anti-Wg (DSHB 1:50), mouse anti-Repo (DSHB 1:50), mouse anti-Fz1 (DSHB 1:50), Rabbit anti-Fz1 [139], 1:300), mouse anti-Cyt-Arm (DSHB 1:50), mouse anti-MMP1 (DSHB 5H7B11, 3A6B4, 3B8D12, 1:50), rabbit anti-MMP2 (1:500, K. Broadie) [140], mouse anti-β-galactosidase (Sigma, 1:500), rabbit anti-GFP (Invitrogen A11122, 1:500), mouse anti-GFP (Invitrogen A11120, 1:500), mouse anti-Nc82(brp) (DSHB 1:20), mouse anti-ELAV (DSHB 1:50), Rabbit anti-Hrp (Jackson Immunoresearch 111-035-144, 1:400).

Secondary antibodies: anti-mouse Alexa 488, 568, 647, anti-rabbit Alexa 488, 568, 647 (Thermofisher, 1:500). DNA was stained with 2-(4-amidinophenyl)-1H-indole-6-carboxamidine (DAPI, 1 μM).

### Cell culture, fixation, and histology of S24 xenograft model

The S24 cell line was derived as a primary glioblastoma culture (Lemke and colleagues, 2012; Osswald and colleagues, 2015). For the S24 glioma model, 50.000 S24:GFP cells (stably

transduced by lentivirus) were injected into the cortex in 8- to 10-week-old male NMRI nude mice (Charles River, Sulzfeld, Germany, $n = 2$). Cells were cultivated under serum-free conditions in DMEM-F12 as sphere cultures (Thermo Fisher Scientific Inc., Waltham, MA) supplemented with 2% B-27 (Thermo Fisher Scientific Inc., Waltham, MA), 5 μg/ml human insulin (Sigma-Aldrich Corporation, St. Louis, MO), 12.8 ng/ml heparin (Sigma-Aldrich), 0.4 ng/ml EGF (R&D Systems Inc., Minneapolis, MN), and 0.4 ng/ml FGF (Thermo Fisher Scientific Inc., Waltham, MA). Animals were sacrificed 90 days after glioma cell injection with age-matched WT NMRI nude mice ($n = 2$), which were used as control (as previously described in the work by Osswald and colleagues [7]). To validate our data, we performed a series of more diffuse tumor parts (history grade II-like tumor periphery in which the normal brain has just been colonized) and denser tumor parts (grade III-IV like, from central areas) GB images from S24 xenografts brain sections.

Brains were fixed with transcardial perfusion with 40 ml PBS and 40 ml 4% PFA. The brain was removed and postfixed in 4% PFA for 4 h at room temperature. Afterward the brains were stored in PBS at 4 ˚C in the dark.

For histology, S24:GFP tumor-bearing brains were coronally cut on a vibratome (Sigmann Elektronik, Hüffenhardt, Germany) into 100 μm sections. The sections were permeabilized with 1% TX100 for 3 h and counterstained with primary antibodies against beta-catenin (abcam, ab32572) and WNT1 (abcam, ab15251) for 3 h in 0.2% TX100 and 5% FCS. Sections were washed 3 times with 0.2% TX100 and 5% FCS and counterstained with secondary antibodies couples to Alexa-647 and Alexa-546 (Invitrogen) as well as DAPI for 3 h. The sections were washed 3 times in 1× PBS (pH 7.4), and mounted on coverslips using self-made moviol. Images were acquired on a confocal laser-scanning microscope (Leica SP8, Leica, Germany) using a ×63 immersion oil objective (numerical aperture = 1.4). z-Stacks were acquired with a pixel size of 141 nm and 300-nm z-steps.

All animal experiments were approved by the regional animal welfare committee (permit number: G132/16 Regierungspräsidium Karlsruhe).

## Western blots

For western blots, we used NuPAGE Bis-Tris Gels 4% to 12% (Invitrogen), and the following primary antibodies: mouse anti-Fz1 (DSHB 1:500), mouse anti-MMP1 (DSHB 1:500), and mouse anti-tubulin (1:10,000 Sigma). We used Tubulin as a loading control instead of actin because the TMs are Actin positive and tubulin negative as previously described by Osswald and colleagues [7]. There were 3 biological replicates and Relative Fz1 or MMP1 Average pixel intensity was measured using measurement tool from Image Studio Lite version 5.2 and normalized against Tubulin.

## PLA

We used the DUO92101 Duolink® In Situ Red Starter Kit Mouse/Rabbit with DUO92013 Duolink In Situ Detection Reagents FarRed (Sigma) for the PLA experiments.

The interaction between Wg and Fz1 in Drosophila larval brains was detected in situ accordingly to the instructions of the manufacturer. Briefly, primary antibody incubation against Wg (mouse anti-Wg [DSHB 1:50] and Fz1 [Rabbit anti-Fz1 [139], 1:300]) were applied using the same conditions as immunocytochemistry staining. Duolink secondary antibodies against the primary antibodies were then added. These secondary antibodies were provided as conjugates to oligonucleotides that were able to form a closed circle via base pairing and ligation using Duolink ligation solution when the antibodies were in close proximity [60] at a distance estimated to be <40 nm. The detection of the signals was conducted

by rolling circle amplification using DNA polymerase incorporating fluorescently labeled nucleotides into the amplification products. The resulting positive signals were visualized as bright fluorescent dots, with each dot representing one interaction event. As a negative control, one of the primary antibodies was not added; therefore, no positive signals were obtained from that assay. The tissues were visualized using a confocal microscope system (LEICA TCS SP5).

## In situ hybridizations

Protocol was performed according to the work by Martin and colleagues [141]. Imaginal discs and brains were dissected and fixed in 4% formaldehyde for 20 min at room temperature, washed in PBS-0.1% Tween (PBT), and refixed for 20 min at room temperature with 4% formaldehyde and 0.1% Tween. After 3 washes in PBT, discs were stored at −20 ˚C in hybridization solution (HS; 50% formamide, 5× SSC, 100 μg/ml salmon sperm DNA, 50 μg/ml heparin, and 0.1% Tween). Disc were prehybridized for 2 h at 55 ˚C in HS and hybridized with digoxigenin-labeled RNA probes at 55 ˚C. The probes were previously denaturalized at 80 ˚C for 10 min. After hybridization, discs were washed in HS and PBT and incubated for 2 h at room temperature in a 1:4,000 dilution of anti-DIG antibody (Roche). After incubation, the discs were washed in PBT, and the detection of probes was carried out using NBT and BCIP solution (Roche). The discs were mounted in 70% glycerol. Images were acquired with a Leica DM750 microscope and Leica MC170HD camera and LAS version 4.8 software. The probes were generated from the cDNAs RE026007 (wg) and LD32066 (fz1) from the Expression Sequence Tags (EST) collection of the Berkeley *Drosophila* Genome Project.

## TEM

TEM was performed in CNS of third instar larvae with horseradish peroxidase (HRP) genetically driven to glial cells. Brains were fixed in 4% formaldehyde in PBS for 30 min at room temperature and washed in PBS, followed by an amplification of HRP signal using the ABC kit (Vector Laboratories) at room temperature. After developing with DAB, brains were washed with PBS and fixed with 2% glutaraldehyde, 4% formaldehyde in PBS for 2h at room temperature. After washing in phosphate buffer, the samples were postfixed with OsO4 1% in 0.1 M phosphate buffer, 1% K3[Fe(CN)6] 1 h at 4 ˚C. After washing in dH$_2$O, Brains were incubated with tannic acid in PBS for 1 min at room temperature, then washed in PBS for 5 min and dH$_2$O 2× for 5 min. Then the samples were stained with 2% uranyl acetate in H$_2$O for 1 h at room temperature in darkness followed by 3 washes in H$_2$O2d. Brains dehydrated in ethanol series (30%, 50%, 70%, 95%, 3× 100% 10 min each at 4 ˚C). Infiltration: samples were incubated in EtOH:propylene's OXID (1:1;V.V) for 5 min, propylene's OXID 2× for 10 min, propylene's OXID:Epon (1:1) for 45 min, Epon 100% in agitation for 1 h, and Epon 100% in agitation overnight. Then change to Epon 100% for 2 to 3 h. Finally encapsulate the samples in BEEM capsules and polymerize 48 h at 60 ˚C [142].

## Neurodegeneration studies

To measure neurodegeneration in our fly model, we quantified the number of active zones (synapses) in the NMJ. This well-established system has been used for decades to study neurodegeneration in *Drosophila* [143–146]; the motor neuron soma is located in the central nervous system, but synapse counting can be accurately done in synaptic buttons located in adult or larva muscular wall by using a brp (Nc82) antibody.

## Imaging

Fluorescent labeled samples were mounted in Vectashield mounting media with DAPI (Vector Laboratories) and analyzed by Confocal microscopy (LEICA TCS SP5/SP8). Images were processed using Leica LAS AF Lite and Fiji (Image J 1.50e). Images were assembled using Adobe Photoshop CS5.1.

## Quantifications

Relative Wg, Fz1, Cyt-Arm, TRE-RFP, MMP1, MMP2, WNT1, and βCatenin staining within brains was determined from images taken at the same confocal settings. Average pixel intensity was measured using measurement log tool from Fiji 1.51g and Adobe Photoshop CS5.1. Average pixel intensity was measured in the glial tissue and in the adjacent neuronal tissue ($N < 10$ for each sample in triplicates) and expressed as a Glia/Neuron ratio in all cases but Cyt-Arm that was expressed as Neuron/Glia ratio. Glial network volume was quantified using Imaris surface tool (Imaris 6.3.1 software).

Total average pixel intensity of WNT1 and βCatenin staining within mice brains was measured in the glioma ($N = 6$) and control samples ($N = 6$); to quantify this, single sections were taken from similar z-positions in both control and glioma samples. Glial network volume was quantified using Imaris surface tool (Imaris 6.3.1 software). The number of Proximity ligation assay puncta, Repo+ cells, and the number of synaptic active sites (Nc82+) was quantified by using the spots tool Imaris 6.3.1 software. We selected a minimum size and threshold for the puncta in the control samples of each experiment. Then we applied these conditions to the analysis of each corresponding experimental sample. The Western blot bands were quantified by using the Image Studio Lite 5.2 software.

## Statistical analysis

To analyze and plot the data, we used Microsoft Excel 2013 and GraphPad Prism 6. We performed a D'Agostino and Pearson normality test, and the data found to have a normal distribution were analyzed by a two-tailed $t$ test with Welch correction. In the case of multiple comparisons, we used a one-way ANOVA with Bonferroni post test. The data that did not pass the normality test were subjected to a two-tailed Mann–Whitney U test or, in the case of multiple comparisons, a Kruskal–Wallis test with Dunns post test. Error bars represent SEM; ***$p \leq 0.0001$, **$p \leq 0.001$, *$p \leq 0.01$, ns = nonsignificant.

## Viability assays

Flies were crossed and progeny was raised at 25 ˚C under standard conditions. The number of adult flies emerged from the pupae were counted for each genotype. The number of control flies was considered 100% viability, and all genotypes are represented relative to controls. Experiments were performed in triplicates.

## Survival assays

Males Tub-Gal80; Repo-Gal4 were crossed with males bearing a control construct (UAS–LacZ) or glioma (UAS–PI3Kdp110; UAS-EGFRλ) and raised at 17 ˚C. Progeny bearing a glioma (experimental) or LacZ (control) chromosomes were put at 29 ˚C, and viability was calculated as the percentage of surviving flies with respect to the starting number of flies as follows: viability = observed (number of flies)/starting number of flies × 100. Six independent vials for glioma ($n = 6$) and control ($n = 6$) were analyzed, with each vial with 10 flies.

Males Tub-Gal80; Repo-Gal4 were crossed with females bearing a control construct (UAS–LacZ) or glioma (UAS–PI3Kdp110; UAS-EGFRλ) or glioma + Mmp1-RNAi and glioma + Mmp2-RNAi and raised at 17 ˚C. Progeny bearing a glioma or glioma + Mmp1-RNAi and glioma + Mmp2-RNAi (experimental) or LacZ (control) chromosomes were put at 29 ˚C, and viability was calculated as the percentage of surviving flies with respect to the starting number of flies as follows: viability = observed (number of flies)/starting number of flies × 100. Six independent vials for experimental ($n = 6$) and controls ($n = 6$) were analyzed, with each vial with 10 flies.

## qRT-PCRs

Total RNA was isolated from larvae brains (Trizol, Invitrogen), and cDNAs were synthesized with M-MLV RT (Invitrogen). The following specific probes from Applied Biosystems were used: Wingless Dm01814379_m1 and Frizzled1 Dm01793718_g1; RpL32 Dm02151827_g1 was used as housekeeping.

qRT-PCR was performed using Taqman Gene Expression (Applied Biosystems) using a 7500 Real Time PCR System (Applied Biosystems) with cycling conditions of 95 ˚C for 10 min and 40 cycles of 95 ˚C for 15 s and 55 ˚C for 1 min. Each experimental point was performed with samples from 2 independent crosses and 3 replicates per experimental point, and differences were assessed with a 2-tailed Student $t$ test. Results were normalized using the housekeeping RpL32 and the ΔΔ cycle threshold method and are expressed as the relative change (-fold) of the stimulated group over the control group, which was used as a calibrator. qRT-PCR results were analyzed with 7500 version 2.0.6 software (Applied Biosystems).

Mmp1 and Mmp2 qRT-PCRs were performed using Sybergreen (Applied Biosystem) with the following primers:

Mmp1F 5´-GACCTGACCTTCACCAGGAA,

Mmp1R 5´-GTTCAAAGCCGCGATAGAAG,

Mmp2F 5´-GTCGGGAGTGGGCTACAATA,

Mmp2R 5´-GATGGCTAGCAAAAGGATCG.

## Supporting information

**S1 Fig. TMs enwrap neurons in GB and cytoneme markers co-localize with glioma network.** (A–B) Control and glioma brains from third instar larvae. Glia is labeled with *UAS-Ihog-RFP* (gray or red in the merge) driven by *repo-Gal4* to visualize TMs in glial cells as part of an interconnecting network. The glial network is marked with lifeActin-GFP reporter (gray or green in the merge), and nuclei are marked with DAPI (blue). Imaris 3D reconstructions are shown in panels A´´´–B´´´). (C–G) Glial network is marked with myr-RFP/ihog-RFP (red) and several additional cytoneme markers: (C) lifeact-GFP reporter (green and glial nuclei are marked with Repo, magenta), (D) GMA-GFP (green), (E) GPI-YFP (green), (F) GFP-MLC (green), (G) sqh-GFP (green) in a glioma brain. (H–J) Down-regulation of neuroglian (*nrg-RNAi*) in glioma brains results in defective TMs (I, red) compared with glioma brains (H); glial nuclei are marked by Repo (green), and TM network volume is quantified in panel J. (K–N) Higher magnifications of control brains (K) showing the glial cytonemes (gray or red in merge) compared with the glioma brains in which the TMs overgrow and enwrap neuronal clusters (L). Upon *igl/Gap43* down-regulation, the glial network does not overgrow or enwrap

neuronal clusters (M) and shows a pattern and size similar to the control. Arrows indicate glial cytonemes/TMs. (N) A viability assay shows that the lethality induced by the glioma is fully rescued upon knockdown of *Gap43/igl*. Nuclei are marked with DAPI (blue). Scale bar size are indicated in this and all figures. The data underlying this figure can be found in S1 Data. Genotypes: (A) *w; lifeActin-GFP; repo-Gal4, UAS-ihog-RFP/UAS-lacZ*, (B-C) *UAS-dEGFR$^\lambda$, UAS-dp110$^{CAAX}$; lifeActin-GFP; repo-Gal4, UAS-ihog-RFP*, (D) *UAS-dEGFR$^\lambda$, UAS-dp110$^{CAAX}$; UAS-GMA-GFP; repo-Gal4, UAS-ihog-RFP*, (E) *UAS-dEGFR$^\lambda$, UAS-dp110$^{CAAX}$; UAS-GPI-YFP/Gal80$^{ts}$; repo-Gal4, UAS-myrRFP*, (F) *UAS-dEGFR$^\lambda$, UAS-dp110$^{CAAX}$; Gal80$^{ts}$; repo-Gal4, UAS-myrRFP/ UAS-GFP-sls(MLC)*, (G) *UAS-dEGFR$^\lambda$, UAS-dp110$^{CAAX}$; Gal80$^{ts}$; repo-Gal4, UAS-myrRFP/ Sqh-GFP*, (H) *UAS-dEGFR$^\lambda$, UAS-dp110$^{CAAX}$; UAS-lacZ; repo-Gal4, UAS-ihog-RFP*, (I) *UAS-dEGFR$^\lambda$, UAS-dp110$^{CAAX}$; UAS-nrg-RNAi; repo-Gal4, UAS-ihog-RFP*, (K) *w; repo-Gal4, UAS-ihog-RFP/UAS-lacZ*, (L) *UAS-dEGFR$^\lambda$, UAS-dp110$^{CAAX}$;; repo-Gal4, UAS-ihog-RFP*, (M) *UAS-dEGFR$^\lambda$, UAS-dp110$^{CAAX}$;; repo-Gal4, UAS-ihog-RFP /UAS-igl-RNAi*. GB, Glioblastoma; GFP, green fluorescent protein; GPI, glycosylphosphatidylinositol; ihog, interference hedgehog; myr, myristoilated; RFP, Red Fluorescent Protein; sqh, spaghetti squash; TM, tumor microtube; YFP, yellow fluorescent protein.
(TIF)

**S2 Fig. igl/*Gap43* knockdown does not show effects in the number of synapses in the NMJ, in the glial network, or in the viability of the flies.** (A–B) Glial network is marked with ihog-RFP (gray or red in the merge). Glial cells are stained with Repo (gray or green in merge), and the number of glial cells are quantified in the following genotypes: *Control*, *Glioma* showing an increase in Repo+ cells, *Glioma;lacZ*, and *Glioma;yellow-RNAi* showing a similar number of Repo+ cells to Glioma alone. (C–D) Upon *igl* knockdown by *RNAi* in normal brains, the glial network (red) is similar to the control. Glial cells are marked by Repo in green. Nuclei are marked by DAPI. (E–F) Neurons (Hrp, magenta) from the larval neuromuscular junction are stained with Nc82 showing the synaptic active sites (green). Upon knockdown of *igl*, the number of synapses marked by Nc82 (green) is similar to the control. (F) Graph showing the quantification of the synapse number. (G) A viability assay shows that the knockdown of *igl* does not alter the percent of viability of male and female flies. Error bars show SD; ***$P < 0.0001$ or ns for nonsignificant. The data underlying this figure can be found in S1 Data. Genotypes: (A) *UAS-dEGFR$^\lambda$, UAS-dp110$^{CAAX}$; UAS-yellow-RNAi; repo-Gal4, UAS-ihog-RFP*, (B) 1. *w; repo-Gal4, ihog-RFP/UAS-lacZ* 2. *UAS-dEGFR$^\lambda$, UAS-dp110$^{CAAX}$; repo-Gal4, UAS-ihog-RFP* 3. *UAS-dEGFR$^\lambda$, UAS-dp110$^{CAAX}$; UAS-lacZ; repo-Gal4, UAS-ihog-RFP* 4. *UAS-dEGFR$^\lambda$, UAS-dp110$^{CAAX}$; UAS-yellow-RNAi; repo-Gal4, UAS-ihog-RFP*, (C-D) *w; repo-Gal4, UAS-ihog-RFP/ UAS-igl-RNAi*, (E-F) 1. *w; UAS-CD8-GFP; D42-Gal4/UAS-igl-RNAi* 2. *w; UAS-CD8-GFP; D42-Gal4/UAS-lacZ*, (G) 1. *w; repo-Gal4, UAS-ihog-RFP/UAS-lacZ* 2. *w; repo-Gal4, UAS-ihog-RFP/UAS-igl-RNAi*. Hrp, horseradish peroxidase; igl, igloo; ihog, interference hedgehog; NMJ, neuromuscular junction; RFP, red fluorescent protein.
(TIF)

**S3 Fig. Cases of human GB patients with mutations in WNT or FZD.** Complete analysis of mutations in human GB samples from COSMIC database http://cancer.sanger.ac.uk/cosmic (A) and TCGA databases through the Xena Functional Genomics Explorer (https://xenabrowser.net/) for transcriptional targets of WNT pathway (B), WNT ligands (C), and FZD receptors (D); data are represented in percentage out of 902 or 922 samples. The total number of cases with mutations in any WNT or FZD gene is shown in red. Genes from WNT and FZD family without mutations in GBs is shown in the bottom. (A) The COSMIC database revealed no alterations in WNT1, WNT3A, WNT6, or WNT5A genes, and 6 cases of human GB cases showed mutations in WNT2 (0.6%). We did not find any case of a GB patient with a

gain of expression in FZD2 or FZD4, and 4 cases (0.4%) showed a mutation for FZD9. The total number of mutations related with WNT or FZD genes accounts for 5% of the total GB samples analyzed. (B–D) Analysis of expression levels in primary GB and nontumoral tissues; transcriptional targets for WNT pathway (B) are up-regulated in GB samples. WNT ligands from the canonical WNT pathway are not up-regulated (C) and among FZD receptors, only FZD7 shows significant changes in GB tissue (D). FZD, Frizzled; GB, Glioblastoma; TCGA, The Cancer Genome Atlas; WNT, wingless-related integration site.
(TIF)

**S4 Fig. Wg and Fz1 transcription levels are similar between controls and gliomas.** (A) qPCRs with *RNA* extracted from control and glioma larvae showing no change in the transcription (*mRNA* levels) of *wg* or *fz1*. (B) Western blot of samples extracted from control and glioma larvae showing no change in the amount of Fz1 protein. Error bars show SD, ns for nonsignificant. (C) In situ hybridization experiments for Wg and Fz1 in controls and gliomas showing no change in the transcription (*mRNA* levels) of *wg* or *fz1*. *The data underlying this figure can be found in* S1 Data. *Genotypes:* (A-C) 1. *w;; repo-Gal4, ihog-RFP/UAS-lacZ* 2. *UAS-dEGFR$^\lambda$, UAS-dp110$^{CAAX}$;; repo-Gal4, UAS-ihog-RFP.* Fz1, Frizzled1; qPCR, quantitative Polimerase Chain Reaction; Wg, wingless.
(TIF)

**S5 Fig. Wg signaling pathway is active in glioma cells.** (A) Larval brain sections with glial network labeled in gray (red in the merge) and stained with Cyt-Arm (gray or green in the merge). Knockdown of *Fz1* in glioma brains showing a homogeneous Cyt-Arm distribution similar to the control. Quantification of Cyt-Arm staining ratio between Ihog+ and Ihog$^-$ domains is shown in principle Fig 5D. (B–G) Glial cell bodies and membranes are labeled with myrRFP or ihog-RFP (red) driven by *repo-Gal4*. Wg signaling pathway reporters *tsh-lacZ* stained with anti-bGal (green) (B–C), *fz4-GFP* in green (D–E), and *dally-lacZ* stained with anti-bGal (green). (C, E, G) Activation of the Wg pathway reporters in GB cells. Genotypes: (A) *UAS-dEGFR$^\lambda$, UAS-dp110$^{CAAX}$; UAS-Fz1-RNAi; repo-Gal4, UAS-ihog-RFP,* (B) *w;; repo-Gal4, UAS-myrRFP/tsh-lacZ,* (C) *UAS-dEGFR$^\lambda$, UAS-dp110$^{CAAX}$;; repo-Gal4, UAS-myrRFP/ tsh-lacZ,* (D) *w;; repo-Gal4, UAS-myrRFP/ fz4-GFP,* (E) *UAS-dEGFR$^\lambda$, UAS-dp110$^{CAAX}$;; repo-Gal4, UAS-myrRFP/ fz4-GFP,* (F) *w;; repo-Gal4, UAS-myrRFP/ dally-lacZ,* (G) *UAS-dEGFR$^\lambda$, UAS-dp110$^{CAAX}$;; repo-Gal4, UAS-myrRFP/ dally-lacZ.* bGal, beta-galactosidase; Cyt-Arm, cytoplasmic-armadillo; Ihog, interference hedgehog; myrRFP, myristoilated red fluorescent protein; Wg, wingless.
(TIF)

**S6 Fig. Wg signaling pathway is active in human glioma cells.** (A–D) A series of grade II and III GB images from S24 xenografts brain sections stained with WNT1 and ß-Catenin (red) show an increase of these signals in grade III when compared with grade II brain sections, indicating that the accumulation of WNT1 and ß-Catenin correlates with the progression of the GB, quantified in panels E–F. (G–H) Technical immunohistofluorescence negative control in NMRI nude control mice brains stained only with the corresponding secondary antibodies showing the background unspecific signal. Nuclei are marked by DAPI (blue). The data underlying this figure can be found in S1 Data. GB, glioblastoma; Wg, wingless; WNT, wingless-related integration site.
(TIF)

**S7 Fig. Glioma network is responsible for the increase in the number of glial cells, and adult *Drosophila* gliomas behave similar to larval gliomas.** (A–D) Larval brain sections with glial cell nuclei stained with Repo (gray). The number of glial cells is quantified in the following

genotypes: (A) Control, (B) Glioma showing an increase in Repo+ cells. (C) Upon knockdown of Fz1 in glioma brains, the number of glial cells is partially restored. (D) Knockdown of igl in glioma cells restores the number of glial cells similar to the control. (E) Quantification of the number of Repo+ cells. (F) Viability assay showing the percental of lethality induced by the glioma that is partially rescued upon knockdown of fz1. (G) Survival curve of adult control or glioma flies after a number of days of glioma induction and progression. (H–N) Adult brain sections 7 days after glioma induction with glial cells are labeled with *UAS-myr-RFP* (gray or red in the merge) to visualize the glial network and stained with Cyt-Arm (gray or green in the merge), Fz1 (gray or blue in the merge), and Wg (gray or green in the merge) antibodies. (H–J) Cyt-Arm staining specifically marks the mushroom, and it is homogeneously distributed in the rest of the brain tissue in control sections and accumulates in the neurons cytoplasm where it is inactive in glioma brains. Quantification of Neuron/Glia Cyt-Arm staining ratio between RFP+ and RFP⁻ domains (J). (H′–I′, K) Fz1 staining show homogeneous localization in the control brains (H′) in blue. In the glioma brains, Fz1 accumulates in the glial trans-formed cells (I′), Glia/Neuron Fz1 average pixel intensity ratio quantification is shown in (K). (L–N) Wg is homogeneously distributed in control brains, with a slight accumulation in the RFP+ structures. Wg accumulates in the glioma network similar to the larval brains. Glia/Neu-ron Wg average pixel intensity ratio quantification is shown in (N). (O) Graph showing syn-apse number quantification of adult NMJs from control flies and glioma-bearing flies. Error bars show SD; ***$P < 0.0001$ or ns for nonsignificant. The data underlying this figure can be found in S1 Data. Genotypes: (A) *w;; repo-Gal4, ihog-RFP/UAS-lacZ*, (B) *UAS-dEGFR$^\lambda$, UAS-dp110$^{CAAX}$;; repo-Gal4, UAS-ihog-RFP*, (C) *UAS-dEGFR$^\lambda$, UAS-dp110$^{CAAX}$; UAS-Fz1-RNAi; repo-Gal4, UAS-ihog-RFP*, (D) *UAS-dEGFR$^\lambda$, UAS-dp110$^{CAAX}$;; repo-Gal4, UAS-ihog-RFP /UAS-igl-RNAi*. (F) 1. *w;; repo-Gal4, ihog-RFP/UAS-lacZ* 2. *w; UAS-fz1-RNAi; repo-Gal4, UAS-ihog-RFP* 3.*UAS-dEGFR$^\lambda$, UAS-dp110$^{CAAX}$;; repo-Gal4, UAS-ihog-RFP* 4. *UAS-dEGFR$^\lambda$, UAS-dp110$^{CAAX}$; UAS-fz1-RNAi; repo-Gal4, UAS-ihog-RFP*, (H, L) *w; Gal80$^{ts}$; repo-Gal4, UAS-myrRFP/UAS-lacZ*, (I, M) *UAS-dEGFR$^\lambda$, UAS-dp110$^{CAAX}$; Gal80ts; repo-Gal4, UAS-myrRFP*. Cyt-Arm, cytoplasmic-armadillo; fz1, frizzled1; igl, igloo; NMJ, nueromuscular junction; RFP, red fluorescent protein; Wg, wingless.
(TIF)

**S8 Fig. Restoration of the glia-neuron Wg/Fz1 signaling equilibrium inhibits glioma pro-gression.** (A–B) Larval brain sections with glial network labeled with *UAS-Ihog-RFP* (gray or red in the merge) and stained with Fz1 (gray or blue in the merge). Nuclei are marked with DAPI (green). (C–D) Larval brain sections with glial network labeled with *UAS-Ihog-RFP* (gray or red in the merge). Neurons are labeled with *lexAop-CD8-GFP* (green) driven by *elav-lexA*. Fz1 overexpression in neurons restore homogeneous Fz1 protein distribution (gray or blue in the merge) in the brain, rescue brain size, and neuron distribution (panel C and magni-fication in panel D) compared to panel A and magnification in panel B where the *elav-lexA* is not present in the glioma brains. Arrows indicate Fz1 staining in the glial membranes at the Glia-neuron interphase of glioma brains and its restored localization in panels C–D. (E–H) Brains from third instar larvae displayed at the same scale. Glia is labeled with *UAS-Ihog-RFP* (gray or red in the merge) driven by *repo-Gal4* to visualize active filopodia in glial cells and stained with Wg or Cyt-Arm (gray or green in the merge). Neurons are labeled with *lexAop-CD8-GFP* driven by *elav-lexA* (blue). Fz1 overexpression in neurons restore homogeneous Wg (gray or green in the merge) (F) and Cyt-Arm (H) protein distribution (gray or green in the merge) in the brain, compared with panels E and G where the *elav-lexA* is not present in the glioma brains. Nuclei are marked by DAPI (blue) in panels E and G. Genotypes: (A, C, E, G) *UAS-dEGFR$^\lambda$, UAS-dp110$^{CAAX}$; lexAop-Fz1; repo-Gal4, UAS-ihog-RFP*, (B, D, F, H)

*UAS-dEGFR$^\lambda$, UAS-dp110$^{CAAX}$; lexAop-Fz1/ elav-lexA, lexAop-CD8-GFP; repo-Gal4, UAS-ihog-RFP*. Cyt-Arm, cytoplasmic-armadillo; Fz1, frizzled1; Wg, wingless.
(TIF)

**S9 Fig. JNK mediates GB progression.** Brains from third instar larvae displayed at the same scale and stained with Repo (gray) in the following genotypes (A–E) *control*, *glioma*, *glioma bsk$^{DN}$*, *Glioma egr$^{-/-}$*, and *glioma grnd$^{EXTRA}$* brain sections. The number of Repo$^+$ cells is quantified in panel F. Error bars show SD; $^*P < 0.01$, $^{**}P < 0.001$, $^{***}P < 0.0001$, or ns for nonsignificant. Scale bar size are indicated in this and all figures. The data underlying this figure can be found in S1 Data. Genotypes: (A) *repo-Gal4, ihog-RFP/UAS-lacZ*, (B) *UAS-dEGFR$^\lambda$, UAS-dp110$^{CAAX}$;; repo-Gal4, UAS-ihog-RFP*, (C) *UAS-dEGFR$^\lambda$, UAS-dp110$^{CAAX}$;; repo-Gal4, UAS-ihog-RFP/ UAS-bsk$^{DN}$*, (D) *UAS-dEGFR$^\lambda$, UAS-dp110$^{CAAX}$; egr$^-$/egr$^-$; repo-Gal4, UAS-ihog-RFP*, (E) *UAS-dEGFR$^\lambda$, UAS-dp110$^{CAAX}$; UAS-grnd$^{EXTRA}$/repo-Gal4, UAS-ihog-RFP*. GB, glioblastoma; JNK, cJun N-terminal kinase.
(TIF)

**S10 Fig. MMP2 is up-regulated in GB.** Brains from third instar larvae displayed at the same scale. Glia is labeled with *UAS-Ihog-RFP* (red) driven by *repo-Gal4* to visualize active cytonemes/TM structures in glial cells and stained with MMP2 (gray or green in the merge). (A) MMP2 is homogeneously distributed in control sections, with a slight accumulation in the Ihog+ projections (B) MMP2 accumulates in the TMs and specifically in the projections that are in contact with the neuronal clusters. (C) Inhibition of *Gap43* by *RNAi* in glioma brains restores a normal glial network and MMP2 does not accumulate, showing a homogeneous staining along the brain section. (D) Inhibition of *Fz1* by *RNAi* in glioma brains restores a normal MMP2 distribution. MMP2 does not accumulate showing a homogeneous staining along the brain section. Nuclei are marked with DAPI (blue). (E) Quantification of MMP2 staining ratio between ihog$^+$ and ihog$^-$ domains. (F) MMP2-GFP reporter (gray or green in the merge) showing activation in the glioma cell membranes (gray or red in the merge). (G–J) Maximal projections of control and glioma brains with glial cells labeled with *UAS-Ihog-RFP* (red) to visualize the glial network and stained with anti-bGal (gray or green in the merge) to visualize *MMP1-lacZ* (G, I) or *MMP2-lacZ* activity (H–J). (G–H) Control brains show few cells with MMPs transcriptional reporters activated in glial cells (arrowheads). (I–J) In glioma brains, there is a similar activation of MMPs reporters, only few cells in glioma cells (arrowheads). (K–M) Glial cell bodies and membranes labeled by CD2-GFP (gray or green in the merge) driven by *repo-Gal4* to the glial cells and stained with MMP2 (gray or red in the merge). (K) MMP2 is homogeneously distributed in control sections. (L) MMP2 accumulates in the glial cells upon Fz1 overexpression. (M) Quantification of MMP2 staining ratio between GFP$^+$ and GFP$^-$ domains. Nuclei are marked with DAPI (blue). Error bars show SD; $^*P < 0.01$, $^{**}P < 0.001$, $^{***}P < 0.0001$, or ns for nonsignificant. Scale bar size is indicated in this and all figures. The data underlying this figure can be found in S1 Data. Genotypes: (A) *repo-Gal4, ihog-RFP/UAS-lacZ*, (B) *UAS-dEGFR$^\lambda$, UAS-dp110$^{CAAX}$;; repo-Gal4, UAS-ihog-RFP*, (C) *UAS-dEGFR$^\lambda$, UAS-dp110$^{CAAX}$;; repo-Gal4, UAS-ihog-RFP /UAS-Gap43-RNAi*, (D) *UAS-dEGFR$^\lambda$, UAS-dp110$^{CAAX}$; /UAS-Fz1-RNAi; repo-Gal4, UAS-ihog-RFP*, (F) *Gal80$^{ts}$/UAS-dEGFR$^\lambda$, UAS-dp110$^{CAAX}$; MMP2-GFP; repo-Gal4, myr-RFP*, (G) *MMP1-lacZ; repo-Gal4, ihog-RFP/UAS-lacZ*, (H) *MMP2-lacZ; repo-Gal4, ihog-RFP/UAS-lacZ*, (I) *UAS-dEGFRλ, UAS-dp110$^{CAAX}$; MMP1-lacZ; repo-Gal4, UAS-ihog-RFP*, (J) *UAS-dEGFRλ, UAS-dp110$^{CAAX}$; MMP2-lacZ; repo-Gal4, UAS-ihog-RFP*, (K) *w;; Repo-LexA, LexAop-CD2-GFP/CyO*, (L) *w;; Repo-LexA, LexAop-CD2-GFP/LexAop-Fz1*. bGal, beta-galactosidase; GB, glioblastoma; GFP, green fluorescent protein; Ihog, interference hedgehog; MMP,

matrix metalloproteinase; TM, tumor microtube.
(TIF)

**S11 Fig. Wg accumulation in TMs, activation of wg signaling pathway, and glial cell number increase in GB neurodegeneration requires MMPs.** Brains from third instar larvae displayed at the same scale. Glia are labeled with *UAS-Ihog-RFP* (red) driven by *repo-Gal4* to visualize active cytonemes/TM structures in glial cells and stained with Wg (A–D) or Cyt-Arm (F–I) in gray (green in the merge) in the following genotypes *control*, *glioma*, *glioma MMP1-RNAi*, and *glioma MMP2-RNAi* brain sections. (E) Quantification of Wg average pixel intensity staining ratio between ihog$^+$ and ihog$^-$ domains. (J) Quantification of Cyt-Arm average pixel intensity neuron/glia ratio between ihog$^-$ and ihog$^+$ domains. Nuclei are marked with DAPI (blue). (K–P) Neurons from the larval neuromuscular junction are stained with Nc82 (brp) showing the synaptic active zones in gray. Upon glioma induction (L), the number of synapses (gray) is reduced when compared with the control (K). The number of synapses is restored upon knockdown of *MMP1* or MMP2 (N, P). The quantification of the number of synaptic active zones in all genotypes is shown in Fig 10J. Error bars show SD; $^*P < 0.01$, $^{**}P < 0.001$, $^{***}P < 0.0001$, or ns for nonsignificant. Scale bar size is indicated in this and all figures. The data underlying this figure can be found in S1 Data. Genotypes: (A, F, K) *repo-Gal4, ihog-RFP/UAS-lacZ*, (B, G, L) *UAS-dEGFR$^\lambda$, UAS-dp110$^{CAAX}$;; repo-Gal4, UAS-ihog-RFP*, (M) *UAS-MMP1-RNAi; repo-Gal4, ihog-RFP*, (C, H, N) *UAS-dEGFR$^\lambda$, UAS-dp110$^{CAAX}$; UAS-MMP1-RNAi; repo-Gal4, UAS-ihog-RFP*, (O) *UAS-MMP2-RNAi; repo-Gal4, ihog-RFP*, (D, I, P) *UAS-dEGFR$^\lambda$, UAS-dp110$^{CAAX}$; UAS-MMP2-RNAi; repo-Gal4, UAS-ihog-RFP*. Cyt-Arm, cytoplasmic-armadillo; GB, glioblastoma; ihog, interference hedgehog; MMP, matrix metalloproteinase; TM, tumor microtube; Wg, wingless.
(TIF)

**S1 Data. The file S1 Data contain all numerical data pertaining to the following figures: Figs 1C, 1Q, 1R, 2C, 2H, 2M, 2N, 3D, 3K, 3N, 4E, 4F, 4J, 5J, 5F, 5N, 6E, 6F, 6M, 7F, 7L, 8C, 8D, 8E, 8H, 9F, 10G, 10H, 10I and 10J; S1J, S1N, S2B, S2F, S2G, S4A, S4B, S6E, S6F, S7E, S7F, S7G, S7J, S7K, S7N, S7O, S9F, S10E, S10M, S11E and S11J Figs.**
(XLSX)

**S1 Video. Control network.** 3D video reconstruction of control brains with glia labeled with *ihog-RFP* (*repo>ihog-RFP*) in red (gray in the 3D reconstruction) to visualize cytoneme structures in glial cells as part of an interconnecting network.
(AVI)

**S2 Video. Glioma TMs.** 3D video reconstruction of glioma brains with glia labeled with *ihog-RFP* (*repo>ihog-RFP*) in red (gray in the 3D reconstruction) to visualize TMs structures in glial cells as part of an interconnecting network. In glioma brains, the TMs expand across the brain and form perineuronal nests. TM, tumor microtube.
(AVI)

**S3 Video. Glioma *igl-RNAi* network.** 3D video reconstruction of Glioma;*igl-RNAi* brains with glia labeled with *ihog-RFP* (*repo>ihog-RFP*) in red (gray in the 3D reconstruction) to visualize TM structures in glial cells as part of an interconnecting network. Upon *igl* down-regulation, the glial network does not overgrow or enwrap neuronal clusters and shows a pattern and size similar to the control.
(AVI)

**S4 Video. Control lifeActin.** 3D video reconstruction of control brains from third instar larvae. Glia is labeled with *UAS-Ihog-RFP* driven by *repo-Gal4* to visualize cytonemes in glial cells

as part of an interconnecting network (red). Glial network is marked with lifeActin-GFP reporter (green), and nuclei are marked with DAPI (blue).
(MP4)

**S5 Video. Glioma lifeActin.** 3D video reconstruction of gliomal brains from third instar larvae. Glia is labeled with *UAS-Ihog-RFP* driven by *repo-Gal4* to visualize TMs in glial cells as part of an interconnecting network (red). Glial network is marked with lifeActin-GFP reporter (green), and nuclei are marked with DAPI (blue). Glial TMs enwrap clusters of neurons in individual GB perineuronal nests. GB, gliblastoma; GFP, green fluorescent protein; TM, tumor microtube.
(MP4)

**S6 Video. Climbing assay.** Video of *Drosophila* adult negative geotaxis behavior analysis (climbing assay) as an indication for possible motor defects associated with neurodegeneration. The results showed symptoms of neurodegeneration in glioma flies (right tube) compared to controls (left tube).
(AVI)

**S1 Raw images. Original images supporting western blot results reported in Fig 8D and S4B Fig.**
(PDF)

## Acknowledgments

We thank Professor Alberto Ferrús, Associate Professor Helena Richardson, Dr. Paco Martín, Dr. Elena Santana, and Patricia Jarabo for critiques of the manuscript and for helpful discussions. Clemencia Cuadrado for fly stocks maintenance. We want to thank JF de Célis and C. Martínez Ostalé for their critical help with in situ hybridizations. We are grateful to R. Read, I. Guerrero, M. Milan, A. Baena-López, E. Martín-Blanco, D. Strutt, J.P. Vincent, C. Klambt, P. Leopold, K. Broadie, the Vienna *Drosophila* Resource Centre, the Bloomington *Drosophila* stock Centre, and the Developmental Studies Hydridoma Bank for supplying fly stocks and antibodies, and FlyBase for its wealth of information. We acknowledge the support of the Confocal Microscopy unit and Molecular Biology unit at the Cajal Institute and the *Drosophila* Transgenesis Unit and the Transmission Electron Microscope unit at CBMSO for their help with this project.

## Author Contributions

**Conceptualization:** Marta Portela, Frank Winkler, Sergio Casas-Tintó.

**Formal analysis:** Marta Portela, Varun Venkataramani, Sergio Casas-Tintó.

**Funding acquisition:** Sergio Casas-Tintó.

**Investigation:** Marta Portela, Varun Venkataramani, Natasha Fahey-Lozano, Esther Seco, Maria Losada-Perez, Frank Winkler, Sergio Casas-Tintó.

**Methodology:** Marta Portela, Varun Venkataramani, Maria Losada-Perez, Sergio Casas-Tintó.

**Project administration:** Sergio Casas-Tintó.

**Resources:** Frank Winkler, Sergio Casas-Tintó.

**Supervision:** Frank Winkler, Sergio Casas-Tintó.

**Validation:** Marta Portela, Varun Venkataramani, Sergio Casas-Tintó.

**Visualization:** Marta Portela, Sergio Casas-Tintó.

**Writing – original draft:** Marta Portela, Frank Winkler, Sergio Casas-Tintó.

**Writing – review & editing:** Marta Portela, Frank Winkler, Sergio Casas-Tintó.

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
