## [Editor Report · Decision Letter 0]

9 Jun 2019

Dear Dr Casas Tintó, 

Thank you for submitting the revised version of your manuscript entitled "Glioblastoma cells vampirize WNT from neurons and trigger a JNK/MMP signaling loop that enhances glioblastoma progression and neurodegeneration" for consideration as a Research Article by PLOS Biology.

Your manuscript has now been evaluated by the PLOS Biology editorial staff as well as the original academic editor and I am writing to let you know that we would like to send your submission out for external peer review.

Please re-submit your manuscript within two working days, ie. by Jun 11 2019 11:59PM.

Kind regards,

Ines

--

Ines Alvarez-Garcia, PhD

Senior Editor

PLOS Biology

Carlyle House, Carlyle Road

Cambridge, CB4 3DN

+44 1223–442810

---

## [Decision Letter · Decision Letter 1]

2 Aug 2019

Dear Sergio,

Thank you very much for submitting a revised version of your manuscript "Glioblastoma cells vampirize WNT from neurons and trigger a JNK/MMP signaling loop that enhances glioblastoma progression and neurodegeneration" for consideration as a Research Article at PLOS Biology. I sincerely apologize again for the time it has taken us to provide you with a decision. This revised version of your manuscript has been evaluated by the PLOS Biology editors, and the original Academic Editor and reviewers.

The reviews are attached below. The reviewers are mostly positive about your merged manuscript and the revisions. However, they raised several concerns that you should be able to address mainly by further textual changes. These concerns are, nonetheless, important and should be taken very seriously; particularly, those raised by Reviewer 2. Please note that in fairness to all parties involved in the assessment of your manuscript, including yourself, we will not be able to grant further revision opportunities if the concerns are not addressed satisfactorily. Therefore, you should take on board the reviewers’ criticism and aim to satisfy our referees and to meet our editorial standards in your next submission.

Your revisions should address the specific points made by each reviewer. 

Please submit a file detailing your responses to the editorial requests and a point-by-point response to all of the reviewers' comments that indicates the changes you have made to the manuscript. In addition to a clean copy of the manuscript, please upload a 'track-changes' version of your manuscript that specifies the edits made. This should be uploaded as a "Related" file type. You should also cite any additional relevant literature that has been published since the original submission and mention any additional citations in your response. 

Before you revise your manuscript, please review the following PLOS policy and formatting requirements checklist PDF: http://journals.plos.org/plosbiology/s/file?id=9411/plos-biology-formatting-checklist.pdf. It is helpful if you format your revision according to our requirements - should your paper subsequently be accepted, this will save time at the acceptance stage.

Please note that as a condition of publication PLOS' data policy (http://journals.plos.org/plosbiology/s/data-availability) requires that you make available all data used to draw the conclusions arrived at in your manuscript. If you have not already done so, you must include any data used in your manuscript either in appropriate repositories, within the body of the manuscript, or as supporting information (N.B. this includes any numerical values that were used to generate graphs, histograms etc.). For an example see here: http://www.plosbiology.org/article/info%3Adoi%2F10.1371%2Fjournal.pbio.1001908#s5.

For manuscripts submitted on or after 1st July 2019, we require the original, uncropped and minimally adjusted images supporting all blot and gel results reported in an article's figures or Supporting Information files. We will require these files before a manuscript can be accepted so please prepare them now, if you have not already uploaded them. Please carefully read our guidelines for how to prepare and upload this data: https://journals.plos.org/plosbiology/s/figures#loc-blot-and-gel-reporting-requirements.

Upon resubmission, the editors assess your revision and assuming the editors and Academic Editor feel that the revised manuscript remains appropriate for the journal, we may send the manuscript for re-review. We aim to consult the same Academic Editor and reviewers for revised manuscripts but may consult others if needed.

We expect to receive your revised manuscript within one month. Please email us (plosbiology@plos.org) to discuss this if you have any questions or concerns, or would like to request an extension. At this stage, your manuscript remains formally under active consideration at our journal; please notify us by email if you do not wish to submit a revision and instead wish to pursue publication elsewhere, so that we may end consideration of the manuscript at PLOS Biology.

When you are ready to submit a revised version of your manuscript, please go to https://www.editorialmanager.com/pbiology/ and log in as an Author. Click the link labelled 'Submissions Needing Revision' where you will find your submission record. 

Sincerely,

Ines

--

Ines Alvarez-Garcia, PhD

Senior Editor

PLOS Biology

Carlyle House, Carlyle Road

Cambridge, CB4 3DN

+44 1223–442810

--------------

Reviewers' comments:

Rev. 1:

The authors has follow reviewer indications. As such the merge the manuscripts informations enhance the manuscript focus and the final information to the readers. Still there are sentencees need explanations. My view is that the manuscript need some additional cleaning avoiding sentences with over hestimation of the model. Jumping to the human Glioma is not an easy task.

There are some sentences thougth I found difficult to judge, based on their observations which are not fully addresses, as such, I suggest the authors to deplete or need data to further confirm in this paper . An alternative is to bounce to human, there are now several geneexpression data signature in glioma that can be used to prove in silico this model.

I am not able to judge the fly experiments and the use of all these mutants as presented, I assume they all work perfectly but I am sure some are " leaky mutants "and as such a description of the leakiness should be add in the text in the methology section. Overall there is still some work to be done but the unified version is much improved.

I found some problems on some sentences in the results section then repeated and followed in discussion that need a peculiar attention.

1) "These results suggest that wg expression in glioma cells is not relevant for glioma progression". Althought the experiments are of interest I don't think this in human is the case, persistent WNT signalling is observed in those uncurable tumours, so this cannot be extrapolated from fly to human and as such the mechainism need to be reevaluated.

2)"Altogether, these results reveal a novel positive feedback loop involving Wg, JNK and MMPs that contribute to GB development and progression. These discoveries reveal potential pharmacological targets and novel avenues for the treatment of GB".

Have you consider the crosstalk effects? JNK and MMPs are know to be activated by several other pathways and a crosstalk action of these pathways in human are known to underway to go to a precise dissection.

TGF.beta/BMP/MAPK as well as PI3K has role on activating JNK and MMP mostly via PTEN inhibition, please follow this and comment it into the manuscript.

3) Neurodegeneration because of Wg depletion, ....well this is very strong hypothesis. Neurodegeneration it is supposed to start if there is oxidative stress and mithocondrial dysfunction on the other hand WNT signalling loss is causing synapses neurodegeneration in AD (Alzheimer disease), this os only one example --- so this sentence above need a better rephrasing. Maybe you want to investigate oxidative stress in your model and link to neurodegereration as well as to those initiation events of neurodegeneration such as impairing of the autophagy, but at this time the message goes cunfused becouse too many aspects to take in consideration at this time. My suggestion is cut the story to the major findings.

4)"This study integrates for the first time the oncogenic nature of glioma with the neuronal degeneration caused by Wg depletion" this sentence is not appropriate, see above.

Minor: the vampire picture in the model is not appropriate for this journal. You migth find a better way to represent your "vampirization" hypothesis.

Rev. 2:

I have not been able to read the whole paper. It is formally very disorganized and contain multiple inconsistencies.

Below, some examples:

A crucial part, is the list of references. I have tried to read some of the references but is impossible to access to them. The list of references is totally inconsistent with the refs indicated in the text. If I can't read the papers cited, I can't evaluate the work.

A couple of examples:

1- The authors write: WNT signaling has long been suggested as a hallmark in gliomagenesis associated with the proliferation of stem-like cells in human GBs [48]. The reference 48 in the references sections corresponds to “Cytoneme-mediated contact-dependent transport of the Drosophila decapentaplegic signaling protein.” Roy S, Huang H, Liu S, Kornberg TB. (PMID: PMID: 24385607), which has nothing to do with WNT signaling and gliomagenesis.

2- The mechanisms of Wg delivery are currently under debate. This protein was initially described as a secreted protein [61]. Recent studies have proved that Wg secretion is not necessary for Drosophila development [62].

In the references part:

61. Scherer HJ. A Critical Review: The Pathology of Cerebral Gliomas. J Neurol Psychiatry. 1940;3(2):147-77. Epub 1940/04/01. PubMed PMID: 21610973; PubMed Central PMCID: PMC1088179.

62. Boyle M, Bonini N, DiNardo S. Expression and function of clift in the development of somatic gonadal precursors within the Drosophila mesoderm. Development. 1997;124(5):971- 82. Epub 1997/03/01. PubMed PMID: 9056773.

“To determine if TM-like processes expand as a consequence of the increase in the number of GB cells, we quantified the volume of the TM-like processes and divided by the number of glial cells (Figure 1R). The results show that TM-like processes volume/glial cell number ratio is higher in GB and therefore we conclude that TM-like processes expand in GB.”

The authors aimed to determine whether “TM-like processes expand as a consequence of the increase in the number of GB cells”. At the end of the paragraph they conclude, “we conclude that TM-like processes expand in GB”. But this does not answer the question they opened in the beginning of the paragraph: “if TM-like processes expand as a consequence of the increase in the number of GB cells” It is therefore very difficult to follow and understand.

Fig S1. A, B, show panels labeled at the top if the Fig as

• LifeactinGFP (green) and the panels below show black and white images

• Repo-Ihog-RFP (red) and the panels below show black and white images

• DAPI (blue) and the panels below show merged image.

Why? Again, this inconsistencies in the format make the manuscript very difficult to follow. The Figures should be self-explanatory and easy to digest, and this is not the case in this work.

In the text: “Moreover, using methods that were used to prevent epithelial cytoneme formation (by downregulating neuroglian (nrg-RNAi)) [46], we found that the TM-like processes in GB were also reduced (Figure S1 H-I’).” Those panels show two examples of Glioma; NrgRNAi (same genotype in both of them).The authors claim TM are reduced. Reduced when compared to what? Where is the control to compare with? If they claim they are reduced, this should be quantified. The authors should show these data in a more rigorous manner and provide always controls that allow to, at glance, compare with the experimental conditions. Without this, the conclusions are not supported by the results shown.

The writing and grammar is, as in the previous version, very poor. See, eg: Ihog (Interference hedgehog), which is a type 1 membrane protein shown to mediate the response to the active Hedgehog (Hh) protein signal [42], which accumulates in the epithelial cytonemes [43,44].

As the previous version, this manuscript is below the basic standars required for publication in Plos Biol.

Rev. 3:

The revised manuscript “Glioblastoma cells vampirize WNT from neurons and trigger a JNK/MMP signaling loop that enhances glioblastoma progression and neurodegeneration” has now combined two previous manuscripts and re-organized the story, making it more comprehensive. The paper has been improved.

Most of the concerns from the last review have been addressed with proper experiments completed, except for the question #6. I could not agree that unmarked remaining non-glial cells are neurons. There are mixed cell types in tumor microenvironment including endothelial cells, fibroblasts and immune cells. In addition, as the authors demonstrated, neurons are subjected to degeneration and apoptosis. How would they assume that neurons largely existed and were all myr-RFP/Ihog-RFP negative cells? Neuron markers are strongly recommended to show colocalization with Wg reporters.

6. In Fig 4E-H and Fig S7B-G, Wg pathway reporters were only co-stained with glia but not with neuron. How was the conclusion drawn that the reporter activities were down in neurons?

In both figures, control samples (without GB) are included and arrows indicate the activity of Wg reporters in neurons (Figure 3-E, G and Figure S5 B, D and F). Please take into consideration that in the Drosophila model, all glial cells in control brain and glioma were marked with myr-RFP/Ihog-RFP, so the unmarked remaining cells should be neurons.

Overall, the quality of the revised manuscript is improved, and a further revised version is in favor of publication.

---

## [Decision Letter · Decision Letter 2]

15 Oct 2019

Dear Dr Casas Tintó,

Thank you for submitting your revised Research Article entitled "Glioblastoma cells vampirize WNT from neurons and trigger a JNK/MMP signaling loop that enhances glioblastoma progression and neurodegeneration" for publication in PLOS Biology. I have now obtained advice from one of the original reviewers and have discussed their comments with the Academic Editor. 

We're delighted to let you know that we're now editorially satisfied with your manuscript. However before we can formally accept your paper and consider it "in press", we also need to ensure that your article conforms to our guidelines. A member of our team will be in touch shortly with a set of requests. As we can't proceed until these requirements are met, your swift response will help prevent delays to publication.

Early Version

Sincerely,

Ines

--

Ines Alvarez-Garcia, PhD

Senior Editor

PLOS Biology

Carlyle House, Carlyle Road

Cambridge, CB4 3DN

+44 1223–442810

ETHICS STATEMENT:

The Ethics Statements in the submission form and Methods section of your manuscript should match verbatim. Please ensure that any changes are made to both versions.

-- Please include the specific national or international regulations/guidelines to which your animal care and use protocol adhered. Please note that institutional or accreditation organization guidelines (such as AAALAC) do not meet this requirement.

Regardless of the method selected, please ensure that you provide the individual numerical values that underlie the summary data displayed in the following figure panels:

Fig. 1C, Q, R; Fig. 2C, H, M, N; Fig. 3D, K, N; Fig. 4E, F, J; Fig. 5J, F, N; Fig. 6E, F, M; Fig. 7F, L; Fig. 8C, D, E, H; Fig. 9F; Fig. 10G, H, I, J; Fig. S1J, N; Fig. S2B, F, G; Fig. S3B, C, D; Fig. S4A, B; Fig. S6E, F; Fig. S7E, F, G, J, K, N, O; Fig. S9F; Fig. S10E, M and Fig. S11E, J

they are essential for readers to assess your analysis and to reproduce it. Please also ensure that figure legends in your manuscript include information on WHERE THE UNDERLYING DATA CAN BE FOUND.

For manuscripts submitted on or after 1st July 2019, we require the original, uncropped and minimally adjusted images supporting all blot and gel results reported in an article's figures or Supporting Information files. We will require these files before a manuscript can be accepted so please prepare them now, if you have not already uploaded them. Please carefully read our guidelines for how to prepare and upload this data: https://journals.plos.org/plosbiology/s/figures#loc-blot-and-gel-reporting-requirements.

Reviewers' comments

Rev. 3:

The authors have addressed all concerns I raised before.

---

## [Editor Report · Decision Letter 3]

13 Nov 2019

Dear Dr Casas Tintó,

On behalf of my colleagues and the Academic Editor, Jeremy N Rich, I am pleased to inform you that we will be delighted to publish your Research Article in PLOS Biology. 

PRESS 

Since you are planning a press release for this article, your publication date will be scheduled for 17th December 2019.

Kind regards,

Hannah Harwood

Publication Assistant, 

PLOS Biology

on behalf of

Ines Alvarez-Garcia,

Senior Editor

PLOS Biology